# Safe and Stable Control via Lyapunov-Guided Diffusion Models

**Xiaoyuan Cheng**[*]   **Xiaohang Tang**   **Yiming Yang**
University College London, United Kingdom

## Abstract

Diffusion models have made significant strides in recent years, exhibiting strong generalization capabilities in planning and control tasks. However, most diffusion-based policies remain focused on reward maximization or cost minimization, often overlooking critical aspects of safety and stability. In this work, we propose Safe and Stable Diffusion ($S^2$Diff), a model-based diffusion framework that explores how diffusion models can ensure safety and stability from a Lyapunov perspective. We demonstrate that $S^2$Diff eliminates the reliance on both complex gradient-based solvers (e.g., quadratic programming, non-convex solvers) and control-affine structures, leading to globally valid control policies driven by the learned certificate functions. Additionally, we uncover intrinsic connections between diffusion sampling and Almost Lyapunov theory, enabling the use of trajectory-level control policies to learn better certificate functions for safety and stability guarantees. To validate our approach, we conduct experiments on a wide variety of dynamical control systems, where $S^2$Diff consistently outperforms both certificate-based controllers and model-based diffusion baselines in terms of safety, stability, and overall control performance.

## 1 Introduction

Real-world control tasks often go beyond simple cost minimization or reward maximization, requiring safety [19] and stability [7] as essential requirements in various fields such as robotics and aerospace. Specifically, *safety* ensures a risk-free trajectory during control, while *stability* drives the system toward convergence to achieve a desired goal. However, achieving both safety and stability remains an open problem in control theory due to the complexity of synthesizing numerous non-convex constraints, including both equalities and inequalities [21], within a single formulation.

**Model Predictive Control.** A common approach to address this challenge is to leverage convex optimization techniques, minimizing the accumulated costs under multiple constraints [11, 52]. One of the most well-known methods is the model predictive control (MPC) [23] and its variants [17, 18]. While MPC is widely used to ensure safety and stability, the policies derived from MPC algorithms often remain suboptimal in execution [38]. Moreover, MPC can easily become infeasible for some problems with many safety constraints and tight control limitations. Another significant drawback of MPC is its computational complexity [24]. Achieving better performance typically requires a longer receding horizon, making MPC inefficient for high-dimensional nonlinear problems.

**Certificate-Based Method.** Another approach to ensuring safety and stability in control relies on certificate functions, such as control Lyapunov functions (CLFs) [4, 6, 43] and control barrier functions (CBFs) [5, 36, 46]. However, identifying a valid certificate function is challenging and highly problem-dependent [1], as most existing methods rely on convex optimization with known system dynamics [34, 37]. In addition, these approaches struggle to scale to high-dimensional control

---

[*]Corresponding to Xiaoyuan Cheng: `ucesxc4@ucl.ac.uk`

tasks. To improve the generality of certificate-based methods, several approaches have been proposed to learn certificate functions from data [3, 12, 45, 53, 54]. These techniques have been shown effective in control-affine dynamics, enabling the learning of a control policy based on optimized certificate functions. Specifically, the policy is generated by solving step-wise quadratic programs (QPs) using learned parameterized certificate functions and a nominal policy. Two representative methods in this field are proposed in [35] and [13]. The former [35] formulates an approach to synthesizing CBFs for goal-oriented multi-agent control tasks, while the latter [13] introduces the control Lyapunov barrier function (CLBF) which ensures safe and stable control performance [36]. However, synthesizing QP-based control with certificate functions remains challenging [39, 50], due to several factors: (1) QP formulation typically requires control-affine dynamics; (2) ensuring the existence of a control policy for step-wise greedy optimization often requires the introduction of slack variables, which can lead to globally inconsistent behavior; and (3) jointly learning the certificate and policy may lead to an infeasible QP, destabilizing training and degrading certificate quality.

**Contribution.** To address the aforementioned challenges in gradient-based methods, we propose **Safe and Stable Diffusion** ($S^2$**Diff**), a novel sampling-based method based on model-based diffusion planning framework[2], where we learn the certificate functions based on policies sampled via diffusion models and do policy improvement via guided policy sampling iteratively. Crucially, since the guidance is a learned certificate functions (CLBF), the control sequence obtained by guided diffusion sampling is guaranteed to be *safe and stable*. Unlike previous MPC or QP-based methods, our approach goes beyond step-wise greedy policies for control-affine dynamics, avoids the use of slack variables, and can be extended to trajectory-level optimization for general differentiable dynamics (the core idea is illustrated in Figure 1). Theoretically, we uncover intrinsic connections between diffusion sampling and Almost Lyapunov theory: *the diffusion-sampled certificate function is not merely a weak relaxation of classical methods — it constitutes a realization of Almost Lyapunov theory, which prioritizes global convergence rather than pointwise strict descent.*

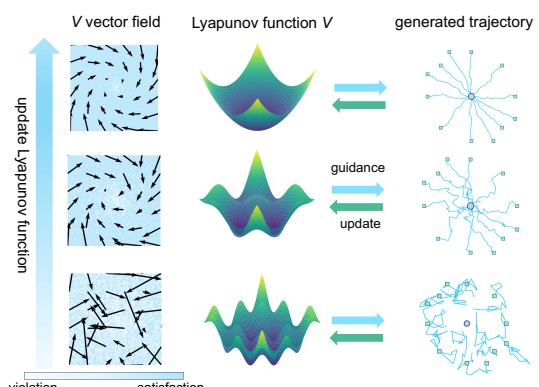

Figure 1: Overview of $S^2$Diff. Bottom to top: as guidance function improves, vector fields align better with the goal, Lyapunov landscapes get smooth, and trajectories converge more reliably. Right to left: the generated policy is guided by the Lyapunov function, and diffusion sampled trajectories are used to update the Lyapunov function.

In experiments, we evaluate $S^2$Diff across a wide range of dynamical systems, where it consistently outperforms both gradient-based methods and model-based diffusion approaches in terms of safety and stability. We summarize the comparison with other methods in Table 1.

**Related Work.** In parallel with gradient-based approaches, an increasing number of works have explored diffusion models [26, 41, 42] for control problems [2, 15, 20, 29, 30, 47, 49]. One work focuses specifically on diffusion planning, as formulated in Diffuser [22], a probabilistic diffusion model that generates action plans by iteratively denoising trajectories. Diffusion planning shows strong performance in long-horizon decision-making and trajectory generation at test time [22, 30]. To address safety in control, model-free safe diffusion planning methods have been proposed, integrating QP based on handcrafted control barrier functions (CBFs) [51]. However, designing valid CBFs prior remains a challenge [1], and the stability guarantee of diffusion policies are still unexplored.

## 2 Preliminary

Consider a general differentiable nonlinear dynamical system given by

$$\dot{x} = f(x, u), \tag{1}$$

---

[2]Model-based diffusion planning here refers to planning with known dynamics as defined in [32].

Table 1: Comparison of existing methods and $S^2$Diff (ours). Policy inference includes, quadratic programming (QP), non-convex optimization (NCO), and Diffusion Sampling (DS).

| Methods | Control-affine | Trajectory level | Safe | Stable | Policy Inference |
|---|---|---|---|---|---|
| CLBF-QP [13] | ✓ | ✗ | ✓ | ✓ | QP (fast) |
| MPC | ✓ | ✓ | ✗ | ✗ | NCO (slow) |
| MBD [32] | ✗ | ✓ | ✗ | ✗ | DS (fast) |
| $S^2$Diff (ours) | ✗ | ✓ | ✓ | ✓ | DS (fast) |

where the state $x$ and control input $u$ belong to the compact sets $\mathcal{X} \subseteq \mathbb{R}^n$ and $\mathcal{U} \subseteq \mathbb{R}^m$, respectively. The function $f : \mathcal{X} \times \mathcal{U} \to \mathbb{R}^n$ is globally Lipschitz. While most existing works on learning certificate functions assume control-affine dynamics [13, 45, 50], we argue that this assumption is unnecessary in our approach due to diffusion sampling. The symbol $\mathcal{X}_s \subseteq \mathcal{X}$ denotes the safe set and $\mathcal{L}$ denotes the Lie derivative. The control Lyapunov barrier function (CLBF) is defined as follows.

**Definition 2.1** (CLBF [36]). A potential function $V : \mathcal{X} \to \mathbb{R}$ is a CLBF if the function $V$ satisfies following conditions, for some constant $c, \lambda > 0$,

$$\textit{Equilibrium:} \qquad V(x_\star) = 0, \tag{2a}$$

$$\textit{Positivity:} \qquad V(x) > 0, \quad \forall x \in \mathcal{X} \setminus \{x_\star\}, \tag{2b}$$

$$\textit{Safe State:} \qquad V(x) \leq c, \quad \forall x \in \mathcal{X}_s, \tag{2c}$$

$$\textit{Unsafe State:} \qquad V(x) > c, \quad \forall x \in \mathcal{X} \setminus \mathcal{X}_s, \tag{2d}$$

$$\textit{Uniform Dissipation:} \qquad \inf_{u \in \mathcal{U}} \mathcal{L}_f V(x) + \lambda V(x) \leq 0, \quad \forall x \in \mathcal{X} \setminus \{x_\star\}. \tag{2e}$$

*Remark* 2.2. CLBFs are Lyapunov-like functions that simultaneously guarantee a system's *safety* and stability. Under the conditions specified in Equations Equation (2a), Equation (2b), and Equation (2e), there exists a control policy that steers the system toward a unique equilibrium point with monotonic decrease. In general, Equation (2e) implies an exponential stability as $\|x_t - x_\star\| \leq \mathcal{O}(\exp(-\lambda t))$, see proof in Appendix C. In this framework, the sublevel and superlevel sets identify the safe and unsafe regions, respectively. Since the uniform dissipation of $V$ is along the trajectory, the state can always be kept in a safe region. With the aid of CLBFs, we establish a rigorous mathematical framework to enforce safety and stability constraints.

**Problem Definition.** Following the conventional setting, the control problem can be formulated as an optimization problem in a fixed horizon $T$ with CLBF $V(\cdot)$. For $\forall t \in [T]$ and $\forall u_t \in \mathcal{U}$:

$$\begin{aligned} \underset{u_{1:T}}{\arg\min} \quad & \sum_{t=1}^{T} q(x_t, u_t) \\ \text{s.t.} \quad & \underbrace{\dot{x}_t = f(x_t, u_t)}_{\text{constraint of dynamics}}, \quad \underbrace{\mathcal{L}_f V(x_t) + \lambda V(x_t) \leq 0}_{\text{constraint of Lyapunov stability}}, \quad \underbrace{V(x_t) \leq c}_{\text{constraint of safety}} . \end{aligned} \tag{3}$$

Here, thesacly cost function $q : \mathcal{X} \times \mathcal{U} \to \mathbb{R}^+$ is predefined and non-negative. The objective is to minimize the accumulated control cost while satisfying multiple constraints. The constraints include conditions derived from the CLBF and bounded control inputs, ensuring global safety and stability.

**Challenges in Gradient-based Methods.** We briefly revisit gradient-based optimization techniques and its challenges. Traditionally, cost minimization problems for control policies are solved using LQR or MPC approaches, often without incorporating safe or stable constraints. The resulting solution, known as the nominal (or reference) control policy denoted as $u^{\text{nominal}}$. Then, the new cost can be simply transformed to $l^2$ distance with $u^{\text{nominal}}$ as $\|u - u^{\text{nominal}}\|^2$. Under this setting, many methods are proposed to incorporate certificate functions $V(\cdot)$ into the one-step policy optimization formulation as QP [13, 39, 40, 45, 50]: for $\forall t \in [T]$,

$$\underset{u}{\arg\min} \|u - u^{\text{nominal}}\|^2, \quad \text{s.t.} \quad \mathcal{L}_f V + \lambda V \leq 0. \tag{4}$$

A feasible solution may not always exist in Equation (4) if $u$ lying a bounded set $\mathcal{U}$. To make the optimization tractable, the constraint term is usually relaxed by introducing a large slack variable

$z \in \mathbb{R}^+$ and coefficient $\lambda_{\text{penalty}}$ showing as

$$\operatorname*{argmin}_{u,z} \|u - u^{\text{nominal}}\|^2 + \boxed{\lambda_{\text{penalty}} z}, \quad \text{s.t.} \quad \mathcal{L}_f V(x_t) + \lambda V(x_t) \le \boxed{z}. \tag{5}$$

In this approach, the optimized control solution is a trade-off between cost minimization and safety/stability. However, solving the problem in Equation (3) using gradient-based optimization presents three challenges: 1). The construction of QP inherently requires dynamics to be control-affine, extending to general nonlinear systems is non-trivial; 2). The step-wise greedy optimization of $u$ can lead to globally inconsistent behavior across the trajectory, often resulting in excessive reliance on a large slack variable $z$ to satisfy CLBF constraints; 3). The CLBF function $V$ is learned jointly with the optimized control policy $u$, which introduces a coupling issue: as $V$ evolves during training, it alters the feasible region of the QP, potentially destabilizing optimization and degrading the learned certificate. These limitations motivate the exploration of alternative approaches— diffusion guided-sampling. We pose the following research question:

*Can guided diffusion sampling overcome the limitations of gradient-based approaches while ensuring both safety and stability?*

We will answer this question by linking Almost Lyapunov theory and diffusion sampling.

## 3 Method

We propose Safe and Stable Diffusion ($S^2$Diff), a model-based diffusion framework that unifies diffusion sampling with Lyapunov-inspired certificates to address safety and stability in general nonlinear systems. The section is organized as follows: (a) We formulate a CLBF-guided diffusion process for sampling trajectory-level policies without requiring slack variables and control-affine assumptions. (b) We design a loss to iteratively refine CLBFs using diffusion-sampled trajectories. (c) We provide theoretical insights linking diffusion sampling to Almost Lyapunov theory.

### 3.1 Probabilistic Formulation and Diffusion Sampling

To apply diffusion for sampling a safe and stable control policy, we first reformulate Equation (3) as a probabilistic formulation. This formulation captures the key insight of the Almost Lyapunov theorem: even if the Lie derivative condition is locally violated with small probability, as long as such violations are confined to regions with minimal influence, the overall system can still exhibit long-term exponential decay, ensuring global stability and safety. By framing probabilistic formulation as a sampling task, we can leverage diffusion models to efficiently explore safe and stable trajectories.

We define the target trajectory sequence distribution $p(U)$ as a Gibbs measure, proportional to

$$p(U) \propto p_{\text{safe}}(U)\, p_{\text{stable}}(U)\, p_{\text{cost}}(U), \tag{6}$$

where $U$ denotes the full trajectory sequence $(x_{1:T}, u_{1:T})$ over a fixed horizon. Here, $p_{\text{safe}}(U)$, $p_{\text{stable}}(U)$, and $p_{\text{cost}}(U)$ are scalar-valued functions that score the trajectory according to safety, stability, and accumulated cost, respectively. These functions shape the unnormalized distribution over policies, reflecting their relative importance. Leveraging the definition of CLBFs, we specify these components as follows:

$$p_{\text{safe}} \propto \prod_{t=1}^T \mathbb{1}_{\{V(x_t) \le c\}}, \; p_{\text{stable}} \propto \prod_{t=1}^T \mathbb{1}_{\{\mathcal{L}_f V(x_t) + \lambda V(x_t) \le 0\}}, \; p_{\text{cost}} \propto \exp\left(-\frac{1}{\gamma}\sum_{t=1}^T q(x_t, u_t)\right), \quad (7)$$

where the variable notation $U = (x_{1:T}, u_{1:T})$ for each probability is omitted for simplification, $\mathbb{1}_{\{\cdot\}}$ denotes the indicator function. The definitions of $p_{\text{safe}}$ and $p_{\text{stable}}$ adhere to the CLBF conditions as stated in Definition 2.1, while $p_{\text{cost}}$ is expressed as an exponential function of the negative cumulative cost with a temperature parameter $\gamma$. When a nominal control policy $u_{1:T}^{\text{nominal}}$ is given, the cost term can be formulated as $p_{\text{cost}} \propto \exp\left(-\frac{1}{\gamma_1}\|u_{1:T} - u_{1:T}^{\text{nominal}}\|^2\right)$, which biases the sampling process toward the nominal policy and improves efficiency.

**Almost Lyapnunov Function Guidance.** As demonstrated in prior work [10, 16], it is theoretically impossible to learn a Lyapunov function that satisfies the required properties at every point, such as

a strictly negative Lie derivative using only a finite number of sampled data points. Nevertheless, Almost Lyapunov theory [27] suggests that despite the existence of regions with non-negative Lie derivative, system trajectories can still converge to a sufficiently small neighborhood around the equilibrium point. Thus, instead of enforcing the second hard constraint in Equation (7), we can adopt a soft constraint formulation:

$$p_{\text{stable}} \propto \exp\left(-\frac{1}{\gamma_2}\sum_{t=1}^{T}\|\left[\mathcal{L}_f V(x_t) + \lambda V(x_t)\right]^+\|^2\right),\tag{8}$$

where $[z]^+ \triangleq \text{ReLU}(z)$ and $0 < \gamma_2 \ll 1$ is a low temperature parameter. When the temperature factor $\gamma_2$ is sufficiently small, the sampled control policy can guarantee trajectory-level safety and stability under the guidance of CLBF. We provide a statement to verify this point see Theorem 3.1. Moreover, this probabilistic (soft) constraint formulation addresses a key limitation of gradient-based methods, which often require introducing large slack variables that can shift the feasible region of the control solution (see Equation (5)). In contrast, the soft constraint in Equation (8) leaves the optimization problem (3) unchanged and improves sampling efficiency by avoiding the high rejection rate associated with strict indicator-based constraints.

**Control Trajectory Sampling.** We adopt Monte Carlo score ascent within diffusion sampling to iteratively denoise trajectories toward the target distribution $p(U)$. The forward and reverse diffusion process with Gaussian noise is defined as follows [42]. Denote scale factor $\bar{\alpha}_i = \prod_{k=1}^{i}\alpha_k$,

$$p(U^i \mid U^0) \sim \mathcal{N}\left(\sqrt{\bar{\alpha}_i}U^0, (1-\bar{\alpha}_i)I\right), \quad U^{i-1} = \frac{1}{\sqrt{\alpha_i}}\left(U^i + (1-\alpha_i)\nabla_{U^i}\log p(U^i)\right).\tag{9}$$

Notably, the unbiased estimator of score function relies on the posterior expectation $\mathbb{E}_{U^0 \sim p(U^0|U^i)}[U^0]$ estimated from sequential Monte Carlo [14, 33] (with provided proof in Lemma B.5):

$$\nabla_{U^i}\log p(U^i) \approx -\frac{1}{1-\bar{\alpha}_i}\left(U^i - \sqrt{\bar{\alpha}_i}\underbrace{\mathbb{E}_{U^0 \sim p(U^0|U^i)}[U^0]}_{\text{estimate via Monte Carlo}}\right).\tag{10}$$

In contrast to QP-based approaches for solving Equation (5), our sampling-based method avoids introducing slack variables and does not require a control-affine structure. After generating the clean trajectory $U^0$, we implement the control policy $u_{1:T}$ in environments.

## 3.2 CLBF Update via Sampled Trajectories

The process of diffusion sampling and CLBF updating is iterative. After generating control policies through diffusion sampling, we use the sampled trajectories to update the CLBF function iteratively. Let $\mathcal{D}$ denote the dataset of sampled trajectories; the CLBF update is formulated as follows:

$$\underset{\hat{V}}{\arg\min}\,\mathbb{E}_{x_{1:T}\sim\mathcal{D}}\Bigg[\sum_{t=1}^{T}\mathbb{1}_{x_t=x_\star}\|\hat{V}(x_t)\| + [-\hat{V}(x_t)]^+ + \mathbb{1}_{x_t\in\mathcal{X}_s}[\hat{V}(x_t)-c]^+ + \mathbb{1}_{x_t\in\mathcal{X}\setminus\mathcal{X}_s}[c-\hat{V}(x_t)]^+$$

$$+\,\alpha_1[\mathcal{L}_f\hat{V}(x_t) + \lambda\hat{V}(x_t) + \epsilon]^+ + \alpha_2[\hat{V}(x_{t+1}) - \hat{V}(x_t) + \lambda\hat{V}(x_t) + \epsilon]^+\Bigg].\tag{11}$$

Here, $\alpha_1$ and $\alpha_2$ are two tunable parameters. Unlike the quadratic form used in prior work [13], we parameterize the CLBF with a general $N$-layer neural network $\hat{V} = W_N\sigma_{N-1}(W_{N-1}\cdots\sigma_1(W_1x))$, to handle nonconvex constraints. Our experiments show that such flexible parameterization outperforms the quadratic form in problems with nonconvex safe sets (see Section 4.1). Each component targets a fundamental control principle as:

- The first term, $\mathbb{1}_{x_t=x_\star}\|\hat{V}(x_t)\|$, enforces $\hat{V}(x_*) = 0$, as required by Equation (2a).

- The second term, $[-\hat{V}(x_t)]^+$, enforces positivity of the Lyapunov function as Equation (2b).

- The third and fourth terms, $\mathbb{1}_{x_t\in\mathcal{X}_s}[\hat{V}(x_t)-c]^+$ and $\mathbb{1}_{x_t\in\mathcal{X}\setminus\mathcal{X}_s}[c-\hat{V}(x_t)]^+$, encode the sub-level and super-level set conditions, aligning with Equations (2c) and (2d).

- The fifth term, $\sum_{t=1}^{T}[\mathcal{L}_f \hat{V}(x_t) + \lambda \hat{V}(x_t) + \epsilon]^+$, enforces trajectory-level stability using diffusion-sampled control inputs.. The buffer term $\epsilon$ accounts for the small violation as formalized in Theorem C.6 (see Equation (32)). Lie derivatives are computed via automatic differentiation, and minimizing this term reduces violations, enhancing global stability.

- Finally, the sixth term, $\sum_{t=1}^{T}[\hat{V}(x_{t+1}) - \hat{V}(x_t) + \lambda \hat{V}(x_t) + \epsilon]^+$ with $x_{t+1} = x_t + f(x_t, u_t)$, complements the continuous constraint by regulating the discrete-time Lyapunov condition.

Our algorithm leverages model-based diffusion as its foundation, and the implementation details are presented in Appendix E.

**Diffusion Sampling vs. Gradient-based Optimization.** Diffusion sampling, as formulated in Section 3.1, avoids the need for slack variables and does not rely on control-affine assumptions, enabling the effective exploration of complex, nonconvex control policy landscapes [32]. As a result, the generated policies can be used to update CLBFs without altering the original optimization objective. In contrast, QP-based methods rely on carefully tuned relaxations and yield locally greedy solutions that may fail globally. Likewise, nonconvex MPC often gets stuck in local minima, limiting its ability to ensure global stability and safety.

## 3.3 Theoretical Results

In this section, we establish the theoretical guarantees of safety and stability for diffusion-sampled policies under the framework of an Almost Lyapunov function. Furthermore, we introduce an auxiliary theorem from a learning-theoretic perspective to demonstrate that diffusion-sampled policies give rise to an Almost Lyapunov function.

**Theorem 3.1** (Safety and Stability with Almost Sure Guarantees). *Let $\mathcal{X}$ be a compact state space and consider the continuously differentiable dynamical system $f$ in Equation (1). Let $V : \mathcal{X} \to \mathbb{R}^+$ be a smooth positive definite function. Assume that there exist constants $\lambda > 0$ and $\epsilon > 0$, and a connected, non-self-overlapping and measurable set $\Omega \subset \mathcal{X}$ satisfying the small volume $Vol(\Omega) < \epsilon$, such that the following holds:*

*(A) For every $x \in \mathcal{X} \setminus \Omega$, the Lie derivative of $V$ along $f$ satisfies $\min_{u \in \mathcal{U}} \mathcal{L}_f V(x) < -\lambda V(x)$.*

*(B) For $x \in \Omega$, we allow $\mathcal{L}_f V(x) \geq -\lambda V(x)$ without any further restrictions, i.e. no uniform dissipation condition.*

*Then, there exist positive constants $\lambda_1$ and $M$, with $0 < \lambda_1 < \lambda$, such that for any $x_0 \in \mathcal{X}_s \subset \mathcal{X}$, the solution $x_t$ of Equation (1) under the diffusion-sampled policy satisfies, almost surely,*

$$V(x_t) \leq \exp(-\lambda_1 t)V(x_0) + M\epsilon^{\frac{1}{n}}, \quad \forall t \geq 0. \tag{12}$$

*In other words, the influence of the "bad" region $\Omega$ introduces only an additive buffer term of order $\mathcal{O}(\epsilon^{\frac{1}{n}})$, ensuring that the overall decay remains almost exponential over time.*

**Connection to Almost Lyapunov Theory.** Theorem 3.1 implies that the CLBF $V$ is not necessary to decrease at every local point in time, reflecting the probabilistic formulation in Equation (8). Instead, even if $V$ leads to local increases, the long-term trajectory exhibits exponential decay (reflecting the core idea of Almost Lyapunov function). As long as any violation is confined to regions with sufficiently weak influence (i.e., small $\epsilon$), the net behavior of system dynamics guarantees global safety and stability. This suggests a robust strategy is to prioritize global performance while allowing a small chance of local Lie derivative violations. On the other hand, the trajectories generated under the diffusion-sampled policy can be reused to train and improve the neural CLBF $V$, progressively reducing the measure of the violation region. Theorem 3.1 holds when the violation region $\Omega$ has sufficiently small volume. This establishes an end-to-end guarantee: the data-driven learning of $V$ ensures the small-volume condition required by the theorem (see detailed proof in Appendix C).

## 4 Experiment

In this section, we comprehensively evaluate the performance of our model across various nonlinear dynamical systems, covering both tracking and control tasks. In the first subsection, we compare our

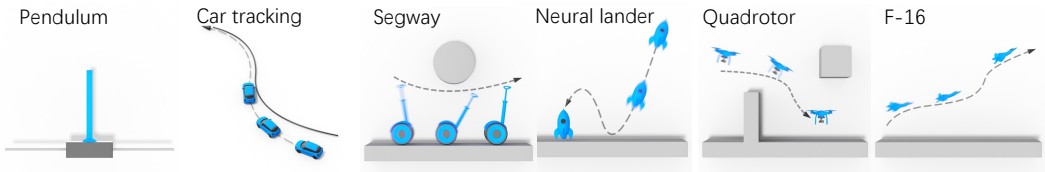

Figure 2: Benchmark control tasks for safety and stability.

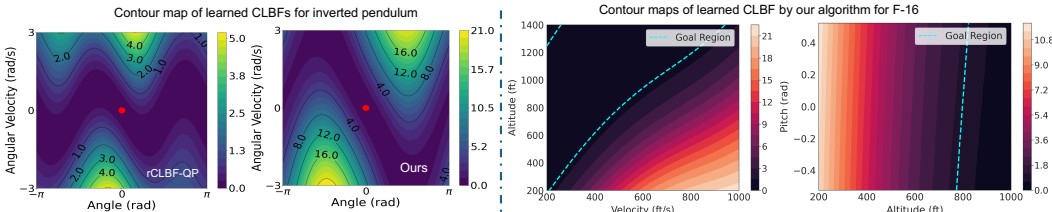

Figure 3: **Left:** CLBFs learned by Gradient-based method (left-1) vs. Diffusion Sampling (left-2) for inverted pendulum . **Right:** Contour maps along different axes of the CLBF learned by $S^2$Diff for the high-dimensional, non-control-affine F-16 with non-convex constraints. The smooth level sets across 2D projections highlight the CLBF's expressiveness and its ability to capture complex, constrained dynamics.

algorithm, $S^2$Diff, with competitive baseline methods and interpret how our design contributes to performance improvements. In the second subsection, we investigate the impact of key hyperparameters, such as sampling horizons and temperature factors, on control performance.

## 4.1 Control Performance Comparison

**Tasks.**  (1) Inverted pendulum ($n = 2$): stabilize the pendulum in the upright position by moving the base left or right. (2) Car trajectory tracking ($n = 5, 7$): make the car follow a desired trajectory by controlling its speed and steering angle. (3) Segway ($n = 4$): keep the two-wheeled robot balanced while moving forward or backward as commanded. (4) Neural lander ($n = 6$): land a aircraft smoothly and accurately on the target in space by adjusting thrusters and orientation. (5) 2D and 3D Quadrotor ($n = 6, 9$): stabilize the quadrotor in hover or level flight with complex obstacles. (6) F-16 ($n = 16$): control a F-16 aircraft to stabilize key variables such as altitude, velocity, angle of attack, elevator position and other critical states. The first five systems are control-affine, while the F-16 is non-control-affine. See illustrations in Figure 2.

**Baseline Algorithms.**   (1) Robust Control Lyapunov Barrier Function based on QP (rCLBF-QP) [13]: a certificate-based control method for control-affine dynamics, utilizing a step-wise greedy policy derived from a learned quadratic CLBF. (2) Model Predictive Control (MPC) [25, 28]: a robust MPC framework designed for safe control tasks. (3) Model-Based Diffusion (MBD) [33]: a model-based diffusion planning approach that leverages known dynamics with constraints.

**Evaluation Metric.**   We evaluate the control policies from three aspects: (1) Safety rate—the percentage of trajectories that remain safe, computed over 20 initial states; (2) Stability—the distance between the fixed-horizon terminal state and the equilibrium state; (3) Efficiency—the inference time required to generate control policies.

**Result Analysis.**   We present an analysis of our results by addressing four key questions that explore the contributions of diffusion sampling and CLBFs in our framework:

(1) *Does diffusion sampling help in learning certificate functions?* Yes—diffusion sampling significantly improves the learning of certificate functions. In Figure 3 (left), we compare contour maps of the CLBF for the inverted pendulum obtained via diffusion sampling and a traditional step-wise QP-based approach. The diffusion-based method results in a noticeably larger contraction region, indicating stronger stability guarantees. Moreover, this advantage generalizes to more complex, high-dimensional systems. For example, in Figure 3 (right), we visualize contour slices of the CLBF

Table 2: Comparison of controller performance in various control environments. The first three tasks are for stability, and other tasks need to consider both safety and stability. − means not applicable.

| Task | Algorithm | Safety rate | $\|x - x_\star\|$ | Eval. time (ms) |
|---|---|---|---|---|
| Inverted Pendulum | rCLBF-QP | – | $0.02 \pm 0.01$ | $\mathbf{4.4 \pm 0.2}$ |
| | MPC | – | $\mathbf{0.01 \pm 0.01}$ | $109.3 \pm 0.8$ |
| | MBD | – | $0.12 \pm 0.05$ | $49.5 \pm 0.3$ |
| | $S^2$Diff | – | $\mathbf{0.01 \pm 0.01}$ | $23.5 \pm 0.1$ |
| Car (Kin.) | rCLBF-QP | – | $0.75 \pm 0.37$ | $\mathbf{10.2 \pm 0.3}$ |
| | MPC | – | $1.51 \pm 0.86$ | $194.6 \pm 1.6$ |
| | MBD | – | $1.57 \pm 0.91$ | $88.5 \pm 1.1$ |
| | $S^2$Diff | – | $\mathbf{0.61 \pm 0.12}$ | $38.2 \pm 0.5$ |
| Car (Slip) | rCLBF-QP | – | $1.03 \pm 0.34$ | $\mathbf{9.6 \pm 0.2}$ |
| | MPC | – | $\mathbf{0.15 \pm 0.09}$ | $336.5 \pm 2.7$ |
| | MBD | – | $1.84 \pm 0.62$ | $69.2 \pm 0.6$ |
| | $S^2$Diff | – | $0.51 \pm 0.18$ | $34.7 \pm 0.9$ |
| Segway | rCLBF-QP | 90% | $\mathbf{0.11 \pm 0.13}$ | $\mathbf{5.2 \pm 0.1}$ |
| | MPC | 20% | $1.39 \pm 0.55$ | $254.6 \pm 4.3$ |
| | MBD | 85% | $1.58 \pm 0.79$ | $205.7 \pm 0.8$ |
| | $S^2$Diff | **100%** | $0.23 \pm 0.09$ | $21.8 \pm 0.3$ |
| Neural Lander | rCLBF-QP | 55% | $0.13 \pm 0.06$ | $\mathbf{12.7 \pm 0.2}$ |
| | MPC | 100% | $0.21 \pm 0.09$ | $247.2 \pm 3.6$ |
| | MBD | 35% | $0.32 \pm 0.19$ | $165.9 \pm 0.4$ |
| | $S^2$Diff | **100%** | $\mathbf{0.06 \pm 0.02}$ | $35.4 \pm 0.7$ |
| 2D Quad | rCLBF-QP | 70% | $0.19 \pm 0.05$ | $\mathbf{18.6 \pm 0.7}$ |
| | MPC | 45% | $0.15 \pm 0.03$ | $279.6 \pm 1.3$ |
| | MBD | 75% | $0.47 \pm 0.29$ | $112.3 \pm 0.2$ |
| | $S^2$Diff | **95%** | $\mathbf{0.11 \pm 0.03}$ | $82.4 \pm 0.3$ |
| 3D Quad | rCLBF-QP | 100% | $0.46 \pm 0.12$ | $\mathbf{9.7 \pm 0.4}$ |
| | MPC | 100% | $0.09 \pm 0.02$ | $324.9 \pm 2.7$ |
| | MBD | 100% | $0.78 \pm 0.38$ | $168.0 \pm 1.6$ |
| | $S^2$Diff | 100% | $\mathbf{0.05 \pm 0.02}$ | $83.5 \pm 0.8$ |
| Average Performance above | rCLBF-QP | 78.75% | 0.384 | **10.06** |
| | MPC | 66.25% | 0.501 | 249.53 |
| | MBD | 73.75% | 0.954 | 122.73 |
| | S$^2$Diff | **98.75%** | **0.226** | 45.64 |
| F-16 (non-control-affine) | rCLBF-QP | – | – | – |
| | MPC | – | – | – |
| | MBD | 100% | $68.34 \pm 32.77$ | $611.3 \pm 13.7$ |
| | $S^2$Diff | **100%** | $\mathbf{47.61 \pm 12.45}$ | $\mathbf{257.2 \pm 5.5}$ |

rCLBF-QP      MPC      MBD      Ours

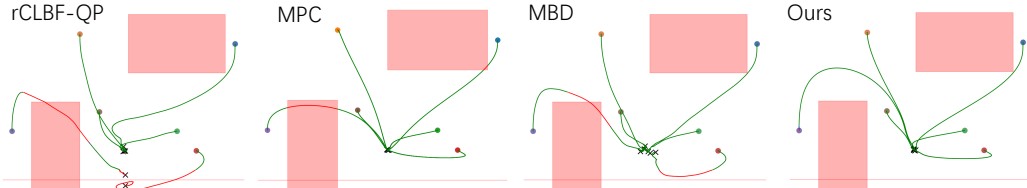

Figure 4: Control trajectories of a 2D quadrotor with four methods including ours ($S^2$Diff). The ○ and × mark the start and end points, respectively. Green lines denote safe states; red lines indicate constraint violations. $S^2$Diff achieves higher safety and stability, effectively handling non-convex constraints where baselines struggle.

learned for the F-16 system. Unlike gradient-based methods reliant on explicit linearization, $S^2$Diff can directly handle general nonlinear and **non-control-affine dynamics**. The resulting non-quadratic level sets highlight the flexibility and expressiveness of the neural network-based CLBF.

(2) *Do diffusion-sampled policies outperform those obtained through gradient-based methods?* Empirical results indicate that diffusion-sampled policies generally achieve superior performance. As shown in Table 2, they outperform both MBD and rCLBF-QP across multiple metrics. This is further illustrated in Figure 4, which depicts trajectories for a 2D quadrotor navigating around non-convex obstacles. While rCLBF-QP performs reasonably well near the goal, the QP-based controller often struggles when the system starts far from the target, due to its myopic nature. In

contrast, diffusion-sampled policies produce smoother, more globally consistent trajectories and maintain a significantly higher safety rate throughout the task.

(3) *Are diffusion-sampled policies further improved under the guidance of CLBFs?* Yes—guiding the diffusion process with a learned CLBF enhances both safety and robustness. Unguided model-based diffusion with fixed penalties offer only local repulsion near unsafe boundaries, leaving most of the space uninformative; as a result, the sampler exhibits low sampling efficiency due to largely unguided exploration, leading to high variance and unsafe outcomes (see Table 2, Figure 5a, 12). In contrast, the CLBF shapes a global energy landscape that funnels trajectories toward safe, goal-directed regions, yielding more stable behavior, consistently safer policies, and lower inference cost.

(4) *Is the Almost Lyapunov theory reflected in our empirical results?* Our empirical evaluations provide strong support for the theoretical guarantees in Theorem 3.1, showing that the violation rate is empirically small (see Table 3). In the F-16 control task, for instance, we monitor violations of the Lie derivative condition and find that they occur with a frequency below $1.5\%$, as illustrated in Figures 5b and 10. These low violation rates, together with the near-monotonic decrease of the CLBF scalar values, indicate that the observed system behavior is consistent with the Almost Lyapunov stability property in Equation (30). Additional quantitative results are provided in Appendix F.2.

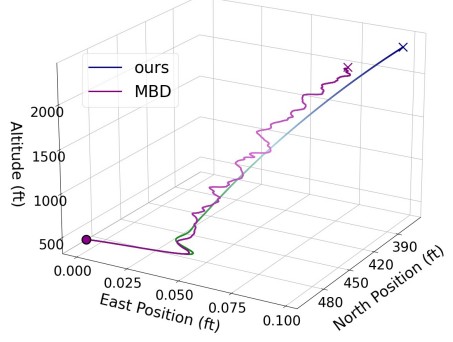

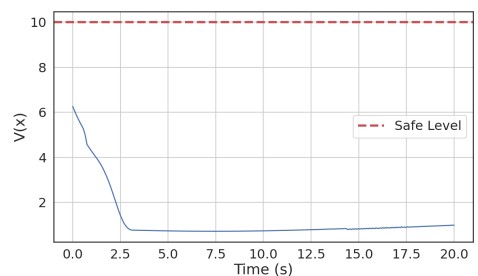

(a) Comparison of control performance between MBD and $S^2$Diff on the F-16 system.

(b) Evolution of the CLBF scalar value under the $S^2$Diff control policy, indicating Almost Lyapunov theory.

Figure 5: Comparison of control performance and Lyapunov behavior on the F-16 system. (a) $S^2$Diff achieves improved tracking compared to MBD. (b) The scalar value of the CLBF under $S^2$Diff shows a nearly monotonic decrease, demonstrating Almost Lyapunov stability.

Table 3: Estimated violation rate $\frac{\text{Vol}(\Omega)}{\text{Vol}(\mathcal{X})}$ on selected benchmark tasks. This fraction quantifies the relative size of the violation region within the compact state set $\mathcal{X}$ (see Equation (18)), and is empirically estimated via uniform sampling over $\mathcal{X}$ to assess the practical violation rate.

| Task | Segway | Neural Lander | 2D Quad | 3D Quad | F-16 |
|---|---|---|---|---|---|
| Violation rate | $0.5\%$ | $1.1\%$ | $1.6\%$ | $1.3\%$ | $2.4\%$ |

## 4.2 Ablation Study

To assess the sensitivity of diffusion sampling and loss function hyperparameters, we conduct a series of ablation experiments in the neural lander environment. More ablation studies in Appendix F.1.

**Violation Temperature.** To evaluate robustness to the stability temperature $\gamma_2$, we train our CLBF with $\gamma_2 \in \{0.5, 0.2, 0.1, 0.05, 0.02, 0.01\}$, fixing $\gamma_1 = 0.5$ (selection from grid search). As shown in Table 4, both high and low $\gamma_2$ degrade performance: high values overly relax stability, increasing violation of stability and safety; low values impose strict constraints, leading to suboptimal solutions. An intermediate value $\gamma_2 = 0.1$ offers the best balance between stability and control performance.

**Hyperparameter in Loss Function.** We study the effect of Lie derivative formulations in the loss (11), comparing automatic differentiation ($\alpha_1$) and discretized approximations ($\alpha_2$). Here, $\alpha_1$

Table 4: Control performance of $S^2$Diff across different temperatures for stability term.

| Metric | 0.5 | 0.2 | 0.1 | 0.05 | 0.02 | 0.01 |
|---|---|---|---|---|---|---|
| Safety rate | 35% | 75% | 100% | 100% | 100% | 100% |
| $\|x - x_\star\|$ | $0.18 \pm 0.08$ | $0.09 \pm 0.06$ | $\mathbf{0.06 \pm 0.02}$ | $0.08 \pm 0.03$ | $0.11 \pm 0.03$ | $0.12 \pm 0.02$ |

and $\alpha_2$ weight the penalties from automatic (auto.) and discretized (dis.) Lie derivatives, respectively. Table 5 shows that using only the discretized term ($\alpha_1 = 0$, $\alpha_2 = 1$) degrades safety and convergence. Incorporating with balanced weights ($\alpha_1 = \alpha_2 = 1$), yields the best stability and accuracy.

Table 5: Control performance of $S^2$Diff under various CLBFs with different $\alpha_1, \alpha_2$ configurations.

| Metric | $\alpha_1 = 0$ $\alpha_2 = 0$ CBF | $\alpha_1 = 1$ $\alpha_2 = 0$ CLBF (auto.) | $\alpha_1 = 0$ $\alpha_2 = 1$ CLBF (dis.) | $\alpha_1 = 1$ $\alpha_2 = 0.5$ - | $\alpha_1 = 0.5$ $\alpha_2 = 1$ - | $\alpha_1 = 1$ $\alpha_2 = 1$ CLBF |
|---|---|---|---|---|---|---|
| Safety rate | 10% | 100% | 85% | 100% | 100% | 100% |
| $\|x - x_\star\|$ | $0.65 \pm 0.14$ | $0.15 \pm 0.07$ | $0.21 \pm 0.09$ | $0.09 \pm 0.03$ | $0.11 \pm 0.05$ | $\mathbf{0.06 \pm 0.02}$ |

## 5   Conclusion

In this work, we introduced Safe and Stable Diffusion ($S^2$Diff), a novel model-based control framework that demonstrates how diffusion sampling can be leveraged to learn certificate functions inspired by Almost Lyapunov theory. Conversely, we show that these learned certificate functions can effectively guide the diffusion process, resulting in safe and stable control policies. Through a probabilistic formulation, our approach avoids the limitations of traditional gradient-based methods, requiring neither relaxations nor control-affine assumptions. Furthermore, the framework is flexible and can be extended to more expressive neural certificate functions, offering a promising direction for safe learning-based control. A current limitation is the slower inference speed compared to QP methods, which may be addressed via policy distillation.

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

# A Notation

| Notations | Meaning |
|---|---|
| $c$ | safety level |
| $f$ | differentiable dynamics |
| $p$ | probability distribution |
| $q$ | cost function |
| $t$ | discrete time step |
| $u$ | control policy |
| $u^{\text{nominal}}$ | nominal control policy |
| $x$ | state |
| $x_\star$ | equilibrium state |
| $\alpha_1, \alpha_2$ | tunable parameters in loss function (11) |
| $\sigma$ | activation function |
| $\gamma, \gamma_1, \gamma_2$ | temperature constants |
| $\lambda$ | dissipation rate |
| $\mathcal{R}$ | Rademacher complexity |
| $U$ | trajectory sequence $(x_{1:T}, u_{1:T})$ |
| $\mathcal{U}$ | compact control policy space |
| $V$ | control Lyapunov barrier function |
| $\mathcal{X}$ | compact state space |
| $\mathcal{X}_s$ | compact safe state space |
| $\mathbb{1}_{\{\cdot\}}$ | indicator function |
| $\mathcal{L}$ | Lie derivative |
| $Vol(\cdot)$ | volume measure function |
| $\Omega$ | violating set lying in the compact set $\mathcal{X}$ |
| $\|\square\|_2$ | spectrum norm of operator |
| $\|\square\|_{2,1}$ | $1-$norm of $2-$norm of the rows of matrices |

# B  Preliminaries

**Definition B.1** (Safe and Stable Control Policy). Given a differentiable dynamical system $\dot{x}_t = f(x_t, u_t)$ with a target goal $x_\star$, a non-empty safe set $\mathcal{X}_s$, and an avoid set $\mathcal{X} \setminus \mathcal{X}_s$, a safe and stable control policy must satisfy the following properties:

- **Stability:** The system state $x_t$ asymptotically approaches $x_\star$ within a small tolerance $\epsilon$, i.e.,

$$\limsup_{t \to \infty} \|x_t - x_\star\| \leq \epsilon.$$

- **Safety:** The system remains within the safe set at all times, i.e.,

$$x_t \in \mathcal{X}_s, \quad \forall t \in \mathbb{N}.$$

*Remark* B.2. The stability ensures the reachability of the dynamical system asymptotically contracting to $x_\star$ while avoiding all unsafe states in $\mathcal{X} \setminus \mathcal{X}_s$. We make a mild assumption that a control policy always exists to satisfy both safety and stability requirements.

**Definition B.3** (Lie Derivative). Let $V : \mathcal{X} \to \mathbb{R}$ be a continuously differentiable scalar function and let $f : \mathcal{X} \times \mathcal{U} \to \mathbb{R}^n$ be a continuously differentiable vector field. The *Lie derivative* of $V$ along the vector field $f$, denoted $\mathcal{L}_f V$, is defined as:

$$\mathcal{L}_f V(x) = \nabla V(x)^\top f(x, u)$$

where $\nabla V(x) \in \mathbb{R}^n$ is the gradient of $V$ at point $x$. This quantity represents the rate of change of $V$ along the trajectories of the dynamical system $\dot{x} = f(x, u)$.

**Proposition B.4.** *Let $V : \mathcal{X} \to \mathbb{R}$ be a continuously differentiable function satisfying the following conditions [6]:*

1. ***Equilibrium:** $V(x_\star) = 0$,*

2. ***Positivity:** $V(x) > 0$ for all $x \in \mathcal{X} \setminus \{x_\star\}$,*

3. ***Uniform dissipation:** There exists $\lambda > 0$ such that*

$$\inf_{u \in \mathcal{U}} \mathcal{L}_f V(x) + \lambda V(x) \leq 0, \quad \forall x \in \mathcal{X} \setminus \{x_\star\},$$

*where $\mathcal{L}_f V(x)$ is the Lie derivative of $V$ along the system dynamics $\dot{x} = f(x, u)$.*

*Then the equilibrium point $x_\star$ is exponentially stable under a suitable control policy $u \in \mathcal{U}$, in the sense that:*

$$V(x_t) \leq V(x_0)e^{-\lambda t}, \quad \forall t \geq 0.$$

*Moreover, if $V(x)$ is radially unbounded and satisfies bounds*

$$\alpha_1(\|x - x_\star\|) \leq V(x) \leq \alpha_2(\|x - x_\star\|),$$

*for some class $\mathcal{K}_\infty$ functions $\alpha_1, \alpha_2$, then:*

$$\|x_t - x_\star\| \leq \beta(\|x_0 - x_\star\|, t),$$

*for some class $\mathcal{KL}$ function $\beta$, and in particular:*

$$\|x_t - x_\star\| \leq Ke^{-\lambda t}, \quad \text{for some } K > 0.$$

While the ideal Lyapunov function highlights that uniform dissipation ensures exponential stability, learning such a perfect function from finite data is theoretically impossible. Rather than focusing on these strict conditions, we turn to the framework of Almost Lyapunov theory.

**Lemma B.5** (Monte-Carlo Estimation of Score). *In the Diffusion process defined as follows. The marginal distribution satisfies:*

$$p(U^0) = \int p(U^N) \prod_{i=1}^{N} p(U^{i-1} \mid U^i) \, dU^{1:N}.$$ (13)

*And the forward process:*

$$p(U^i \mid U^{i-1}) \sim \mathcal{N}(\sqrt{\alpha_i} U^{i-1}, (1 - \alpha_i)I),$$ (14)

*where $\alpha_i$ is the scaling factor and $I$ is the identity matrix. the score function of corrupted samples can be rewritten as*

$$\nabla_{U^i} \log p(U^i) = \mathbb{E}_{U^0 \sim p(U^0 \mid U^i)} \left[ \frac{1}{1 - \bar{\alpha}_i} (U^i - \sqrt{\bar{\alpha}_i} U^0) \right].$$ (15)

*Proof.* We leverage the estimated for Diffusion Sampling, we start with:

$$
\begin{aligned}
\nabla_{U^i} \log p(U^i) = \frac{\nabla_{U^i} p(U^i)}{p(U^i)} &= \frac{\nabla_{U^i} \int p(U^i \mid U^0) p(U^0) \, dU^0}{p(U^i)} \\
&= \int \frac{\nabla_{U^i} p(U^i \mid U^0) p(U^0)}{p(U^i)} \, dU^0 \\
&= \int \frac{\nabla_{U^i} p(U^i \mid U^0)}{p(U^i \mid U^0)} \frac{p(U^i \mid U^0) p(U^0)}{p(U^i)} \, dU^0 \\
&= \int \nabla_{U^i} \log p(U^i \mid U^0) \, p(U^0 \mid U^i) \, dU^0 \\
&= \mathbb{E}_{U^0 \sim p(U^0 \mid U^i)} \left[ \nabla_{U^i} \log p(U^i \mid U^0) \right].
\end{aligned}
$$ (16)

Using the known form of $p(U^i \mid U^0)$ from Equation (9), the score function becomes:

$$\nabla_{U^i} \log p(U^i) = \mathbb{E}_{U^0 \sim p(U^0 \mid U^i)} \left[ \frac{1}{1 - \bar{\alpha}_i} (U^i - \sqrt{\bar{\alpha}_i} U^0) \right].$$ (17)

$\square$

# C   Theoretical Guarantees

This section is divided into two parts:

- Appendix C.1: we give a learning-theoretic analysis showing that one cannot, in general, learn a neural CLBF that enforces uniform dissipation everywhere on a bounded domain. Instead, with sufficiently many samples, the volume (Lebesgue measure) of the "dissipation-violating" region can be driven arbitrarily close to zero.
- Appendix C.2: we further prove that—even when uniform dissipation fails—our diffusion-sampled control policy still achieves almost-sure stability, provided the volume of the dissipation-violating region is sufficiently small.

## C.1   Convergence Analysis of Neural CLBFs

To establish an error bound for the neural CLBF, we consider the probability—under the uniform distribution over a compact set $\mathcal{X} \subset \mathbb{R}^n$—that the Lyapunov dissipation condition is violated. Specifically, we define the violation probability as

$$p_{\mathcal{X}}(\mathcal{L}_f V(x) + \lambda V(x) > 0) = \mathbb{E}_{x \sim \text{Unif}(\mathcal{X})} \left[ \mathbb{1}_{\{\mathcal{L}_f V(x) + \lambda V(x) > 0\}} \right]$$
$$= \frac{Vol\left(\{x \in \mathcal{X} \mid \mathcal{L}_f V(x) + \lambda V(x) > 0\}\right)}{Vol(\mathcal{X})}, \tag{18}$$

where $Vol(\cdot)$ denotes the volume measure (e.g., Lebesgue measure in probability theory) in $\mathbb{R}^n$. Our goal is to minimize the violation region's volume. This expression quantifies the fraction of the domain $\mathcal{X}$ over which the Lie derivative condition fails to hold. It follows that a small expectation corresponds to a small violation region. Throughout we assume the dataset $\mathcal{D}$ is drawn i.i.d. from the uniform distribution over $\mathcal{X}$; therefore training and test distributions coincide.

**Regularity Assumptions.**   Before proceeding with the formal analysis, we introduce a regularity assumption on the candidate CLBF $V$. Specifically, we assume that $V \in \mathcal{V}$, where $\mathcal{V} \subset C^2(\mathbb{R}^n, \mathbb{R})$ denotes the class of continuously differentiable functions that are uniformly bounded in magnitude by a constant $B > 0$; that is $\|f\|_\infty, \|V\|_\infty \leq B$   for all $x \in \mathcal{X}$, where $\mathcal{X} \subset \mathbb{R}^n$ is compact. In addition, we assume that the policy $u(x)$ is Lipschitz continuous with respect to the state $x$, i.e., for all $x_1, x_2 \in \mathcal{X}$, $\|u(x_1) - u(x_2)\| \leq L\|x_1 - x_2\|$ for some constant $L > 0$.

The following lemma provides an upper bound on the probability of constraint violation of uniform dissipation with the learned $V$ in terms of the model complexity and the number of training samples.

**Lemma C.1.** *Fix a $\delta \in (0, 1)$. Assume that the function $V \in \mathcal{V}$ is bounded by a constant $B$ in a compact set. Suppose that the minimization of Equation (18) is feasible and let $V$ denote a solution. The following statement holds with probability at least $1 - \delta$ over the randomness of $x_1, \ldots, x_m$ drawn i.i.d. from dataset $\mathcal{D}$ :*

$$p_{x \sim \mathcal{D}}(\mathcal{L}_f V(x) + \lambda V(x) > -\eta) \leq M\left( \frac{\log^3 m}{\eta^2} \mathcal{R}_m^2(\mathcal{V}) + \frac{2\log(\log(4B/\eta)/\delta)}{m} \right), \tag{19}$$

*where $\mathcal{R}_m(\mathcal{V}) \triangleq \sup_{x_1, \cdots, x_m \in \mathcal{D}} \mathbb{E}_{\epsilon \sim Unif(\{\pm 1\}^m)} \left[ \sup_{V \in \mathcal{V}} \frac{1}{m} \left| \sum_{i=1}^m \epsilon_i (\mathcal{L}_f V(x_i) + \lambda V(x_i)) \right| \right]$ is the Rademacher complexity of the function class $\mathcal{V}$ and $M$ is a universal constant. Also, the positive constant $\eta$ is a margin parameter.*

*Remark* C.2. Lemma C.1 is directly derived from Theorem 5 in [44], which connects the probabilistic error in Equation (18) to an upper bound involving the Rademacher complexity $\mathcal{R}_n(V)$. Different from the previous proof in [10, 16], we prove the general error bound of neural CLBF without any quadratic form assumptions.

**Lemma C.3** (Concentration of the Sampled Control Policy). *Let $u_{1:T}^1, \ldots, u_{1:T}^Q$ be i.i.d. random vectors in $[a, b]^T$ from the target distribution $U$ in Equation (6), and define*

$$u_{1:T}^* = \mathbb{E}[u_{1:T}], \quad \hat{u}_{1:T} = \frac{1}{Q} \sum_{i=1}^Q u_{1:T}^i.$$

*Then for any $\varepsilon > 0$,*

$$p\big(\|\hat{u}_{1:T} - u_{1:T}^*\|_\infty \geq \varepsilon\big) \;\leq\; 2T \exp\Big(-\frac{2Q\varepsilon^2}{(b-a)^2}\Big).$$

*In particular, if*

$$Q \;\geq\; \frac{(b-a)^2}{2\varepsilon^2} \log\Big(\frac{2T}{\delta}\Big),$$

*then with probability at least $1 - \delta$,*

$$\|\hat{u}_{1:T} - u_{1:T}^*\|_\infty \leq \varepsilon.$$

The proof of Lemma C.3 follows by applying Hoeffding's inequality to each coordinate of the trajectory and then taking a union bound over the $T$ time-steps. When the sample number $Q$ is greater than $\frac{(b-a)^2}{2\epsilon^2} \log(\frac{2T}{\delta})$, the probability is at least $1 - \delta$.

To prove the convergence of violation region's volumn of a neural CLBF trained using diffusion-sampled policies, two key factors must be taken into account: (1) the Rademacher complexity of the function class $\mathcal{V}$, denoted $\mathcal{R}(\mathcal{V})$; and (2) the perturbations introduced by the diffusion-sampled policy. In contrast to the noiseless case, we denote the Rademacher complexity of the learned neural CLBF under diffusion-induced noise as $\mathcal{R}(\hat{\mathcal{V}})$.

**Theorem C.4** (Sample Complexity of Neural CLBF). *Under the regularized assumptions, the function class $\mathcal{V}$ is controlled by a $N-$layer neural network with ReLU activation function such that $V(x) = W_N \sigma_{N-1}(W_{N-1} \cdots \sigma_1(W_1 x))$. The following statement holds with probability at least $1 - 2\delta$ over the randomness of $x_1, \ldots, x_m$ drawn i.i.d. from dataset $\mathcal{D}$ and $Q > \frac{(b-a)^2}{2\epsilon^2} \log(\frac{2T}{\delta})$:*

$$p_{x \sim \mathcal{D}}(\mathcal{L}_f V(x) + \lambda V(x) > -\eta)$$

$$\leq M\Bigg[\frac{\log^3 m}{\eta^2} \underbrace{\Bigg(\frac{R}{\sqrt{m}}\big((\|f\|_\infty + \lambda)\Pi_{i=1}^N \|W_i\|_2\big)\Big(\sum_{i=1}^N \big(\frac{\|W_i\|_{2,1}}{\|W_i\|_2}\big)^{2/3}\Big)^{3/2} + \|\nabla_u f\|_\infty \|\Pi_{i=1}^N W_i \epsilon\Bigg)^2}_{\mathcal{R}_m^2(\hat{\mathcal{V}})}$$

$$+ \frac{2\log(\log(4B/\eta)/\delta)}{m}\Bigg].$$

$$(20)$$

*Here, $\|W\|_2$ is the spectral norm of $W$ and $\|W\|_{2,1} \triangleq \sum_l \sqrt{\sum_k W_{l,k}^2}$ is denoted the $1-$norm of $2-$norm of the rows of $W$. Also, $\|f\|_\infty \triangleq \sup_{x \in \mathcal{X}, u \in \mathcal{U}} \|f(x,u)\|_2$, $R \triangleq \sup_{x \in \mathcal{X}} \|x\|_2$.*

*Proof.* We now take $V \in \mathcal{V}$ to be the class of $N-$layer neural network with ReLU activation function:

$$V(x) = W_N \sigma_{N-1}(W_{N-1} \cdots \sigma_1(W_1 x)), \quad \sigma_i(z) = \max\{0, z\}, \tag{21}$$

with bounded spectral norm and row-sum norm $\|W\|_2$ and $\|W\|_{2,1}$, respectively. Bartlett–Foster–Telgarsky [8] show that the Rademacher complexity of $N-$layer neural network with $\rho_i-$lipschitz activation function is

$$\mathcal{R}_m \leq \frac{\Pi_{i=1}^N \rho_i \|W_i\|_2 R}{\sqrt{m}} \Big(\sum_{i=1}^N \big(\frac{\|W_i\|_{2,1}}{\|W_i\|_2}\big)^{2/3}\Big)^{3/2}, \tag{22}$$

where $R \triangleq \sup_{x \in \mathcal{X}} \|x\|_2$.

Back to the Rademacher complexity $\mathcal{R}_m(\hat{\mathcal{V}})$, $\hat{\mathcal{V}}$ is different from $\mathcal{V}$ since it also influenced by perturbations from the diffusion-sampled policy. we first adapt a fundamental fact from the calculus of Rademacher complexities [9], along with the trivial identity

$$\mathcal{R}_m(\hat{\mathcal{V}}) \leq \mathcal{R}_m(\mathcal{V}) + \mathbb{E}_\epsilon\big[\sup_{V \in \mathcal{V}} \frac{1}{m}\big|\sum_{i=1}^m \epsilon\big(\mathcal{L}_{f(x_i,\hat{u}_i)} V(x_i) - \mathcal{L}_{f(x_i,u_i)} V(x_i)\big)\big|\big].$$

Since $|\frac{1}{m}\sum_{i=1}^m \epsilon_i \delta_i| \le \frac{1}{m}\sum_{i=1}^m |\delta_i| \le \sup_i |\delta_i|$, we can further decompose it as

$$\mathcal{R}_m(\mathcal{V}) \le \mathbb{E}_\epsilon\left[\sup_{V\in\mathcal{V}}\frac{1}{m}\big|\sum_{i=1}^m \epsilon_i(\mathcal{L}_f V(x_i) + \lambda V(x_i))\big|\right]$$

$$+ \sup_{x\in\mathcal{X}}\sup_{V\in\mathcal{V}}|\langle f(x_i,\hat{u}_i), \nabla V(x_i)\rangle - \langle f(x_i,u_i), \nabla V(x_i)\rangle|$$

$$\le \underbrace{\mathbb{E}_\epsilon\left[\sup_{V\in\mathcal{V}}\frac{1}{m}\big|\sum_{i=1}^m \epsilon_i(\mathcal{L}_f V(x_i)|\big|\right]}_{\mathcal{R}_1} + \underbrace{\mathbb{E}_\epsilon\left[\sup_{V\in\mathcal{V}}\frac{1}{m}\big|\sum_{i=1}^m \epsilon_i(\lambda V(x_i)|\big|\right]}_{\mathcal{R}_2} \qquad (23)$$

$$+ \underbrace{\sup_{x\in\mathcal{X}}\sup_{V\in\mathcal{V}}|\langle f(x_i,\hat{u}_i), \nabla V(x_i)\rangle - \langle f(x_i,u_i), \nabla V(x_i)\rangle|}_{\Delta_{\text{policy}}}.$$

$\mathcal{R}_2$ is exactly follows the complexity in Equation (22). Based on the definition of Lie derivative, the Rademacher complexity $\mathcal{R}_1$ can be bounded as

$$\mathcal{R}_1 = \mathbb{E}_\epsilon\left[\sup_{V\in\mathcal{V}}\frac{1}{m}\big|\sum_{i=1}^m \epsilon_i(\mathcal{L}_f V(x_i)|\big|\right]$$

$$\le \|f\|_\infty \mathbb{E}_\epsilon\left[\sup_{V\in\mathcal{V}}\frac{1}{m}\big|\sum_{i=1}^m \sup_{\|v_i\|=1}\epsilon_i\langle v_i, \nabla V(x_i)\rangle\big|\right] \qquad (24)$$

$$\le \frac{\|f\|_\infty \Pi_{i=1}^N \rho_i \|W_i\|_2 R}{\sqrt{m}}\left(\sum_{i=1}^N \big(\frac{\|W_i\|_{2,1}}{\|W_i\|_2}\big)^{2/3}\right)^{3/2}.$$

The derivation of Equation (24) from the first to second line is based on Ledoux-Talagrand lemma [48] [3] and Lie derivative properties $|\mathcal{L}_f V(x)| \le \|f\|_\infty \|\nabla V(x)\|_2$. The second line to third line is from the Jacobian Lipschitz property, in which we can use $\Pi_{i=1}^N \rho_i \|W_i\|_2$ to bound the norm of Jacobian matrix. On each linear piece of the ReLU network, the gradient is given by the same weight matrices, so the spectral-norm product likewise controls the complexity of the Jacobian.

The third term $\Delta_{\text{policy}}$ can be bounded according Lemma C.3, and the probability holds with at least $1-\delta$ when sample number $Q > \frac{(b-a)^2}{2\epsilon^2}\log(\frac{2T}{\delta})$:

$$\Delta_{\text{policy}} = \sup_{x\in\mathcal{X}}\sup_{V\in\mathcal{V}}|\langle f(x_i,\hat{u}_i), \nabla V(x_i)\rangle - \langle f(x_i,u_i), \nabla V(x_i)\rangle|$$

$$\le \sup_{x\in\mathcal{X}}\sup_{V\in\mathcal{V}}|\langle f(x_i,\hat{u}_i) - f(x_i,u_i), \nabla V(x_i)\rangle \qquad (26)$$

$$\le \|\nabla_u f\|_\infty \|\nabla V\|_\infty \epsilon \le \|\nabla_u f\|_\infty \Pi_{i=1}^N \rho_i \|W_i\|_2 \epsilon.$$

Due to the properties of ReLU activation function, $\rho_i$ is bounded by 1. Combining the $\mathcal{R}_1$, $\mathcal{R}_2$, Lemma C.1, and perturbations of diffusion-sampled policy in Equation (26) together, we obtain that the probability holds at least $1 - 2\delta$:

$$p_{x\sim\mathcal{D}}(\mathcal{L}_f V(x) + \lambda V(x) > -\eta)$$

$$\le M\left[\frac{\log^3 m}{\eta^2}\left(\frac{R}{\sqrt{m}}\big((\|f\|_\infty + \lambda)\Pi_{i=1}^N\|W_i\|_2\big)\left(\sum_{i=1}^N\big(\frac{\|W_i\|_{2,1}}{\|W_i\|_2}\big)^{2/3}\right)^{3/2} + \|\nabla_u f\|_\infty \Pi_{i=1}^N\|W_i\|_2 \epsilon\right)^2\right.$$

$$\left. + \frac{2\log(\log(4B/\eta)/\delta)}{m}\right]$$

$$(27)$$

Here, $M$ is a universally bounded constant, which avoids to use many constant terms. We can know that the volume of violation region will decrease as $\tilde{\mathcal{O}}(\frac{1}{m})$. The proof end. $\qquad\square$

---

[3]For arbitrary $L-$Lipschitz function $\phi : \mathbb{R}^n \to \mathbb{R}$, we have

$$\mathbb{E}_\epsilon\left[\sup_{g\in\mathcal{G}}\frac{1}{m}\big|\sum_{i=1}^m \epsilon_i(\phi(g(x_i))|\big|\right] \le L\mathbb{E}_\epsilon\left[\sup_{g\in\mathcal{G}}\frac{1}{m}\big|\sum_{i=1}^m \epsilon_i g(x_i)\big|\right]. \qquad (25)$$

## C.2 Almost Sure Guarantees

To establish safety and stability with an almost sure guarantee, we also adopt the regularity assumptions detailed in Appendix C.1. Specifically, we assume that the gradients and Hessians of the candidate CLBF, as well as the dynamics function and its Jacobian, are uniformly bounded over the compact sets $\mathcal{X}$ and $\mathcal{U}$. That is,

$$\|\nabla V\|_\infty, \quad \|\nabla^2 V\|_\infty, \quad \|f\|_\infty, \quad \|\nabla f\|_\infty \leq B$$

for some constant $B > 0$.

**Lemma C.5** (Comparison Lemma [6]). *Suppose that a continuously differentiable function $V \in \mathbb{R}^+ \mapsto \mathbb{R}$ satisfies the following differential inequality:*

$$\dot{V}(x_t) \leq -\lambda V(x_t) + C, \quad \forall t \in \mathbb{R}^+, \tag{28}$$

*where $\lambda \in \mathbb{R}^+$, $C \in \mathbb{R}$, and $\dot{V}(x_0) \in \mathbb{R}$. Then, we have*

$$V(x_t) \leq \exp(-\lambda t)V(x_0) + \frac{C}{\lambda}\big(1 - \exp(-\lambda t)\big). \tag{29}$$

**Theorem C.6** (Safety and Stability with Almost Sure Guarantees). *Let $\mathcal{X}$ be a compact state space and consider the continuously differentiable dynamical system $f$ in Equation (1). Let $V : \mathcal{X} \to \mathbb{R}^+$ be a smooth positive definite function. Assume that there exist constants $\lambda > 0$ and $\epsilon > 0$, and an invariant, connected, non-self-overlapping and measurable set $\Omega \subset \mathcal{X}$ with small volume $Vol(\Omega) < \epsilon$, such that the following holds:*

*(A) For every $x \in \mathcal{X} \setminus \Omega$, the Lie derivative of $V$ along $f$ satisfies $\min_{u \in \mathcal{U}} \mathcal{L}_f V(x) < -\lambda V(x)$.*

*(B) For $x \in \Omega$, we allow $\mathcal{L}_f V(x) \geq -\lambda V(x)$ without any further restrictions, i.e. no uniform dissipation condition.*

*Then, there exist positive constants $\lambda_1$ and $C$, with $0 < \lambda_1 < \lambda$, such that for any $x_0 \in \mathcal{X}_s \subset \mathcal{X}$, the solution $x_t$ of Equation (1) under the diffusion-sampled policy satisfies, almost surely,*

$$V(x_t) \leq \exp(-\lambda_1 t)V(x_0) + C\epsilon^{\frac{1}{n}}, \quad \forall t \geq 0. \tag{30}$$

*In other words, the influence of the "bad" region $\Omega$ introduces only an additive term of order $\mathcal{O}(\epsilon^{\frac{1}{n}})$, ensuring that the overall decay remains* almost *exponential over time.*

*Proof.* We split the argument into two main parts. In Step 1 we show that, by choosing a small "buffer" around the bad set, we recover a uniform dissipation everywhere else. In Step 2 we discretize the trajectory generated by diffusion policy, build a suitable supermartingale, and invoke Azuma–Hoeffding together with the Borel–Cantelli lemma to conclude almost-sure exponential decay.

**Step 1.** Uniform dissipation outside a small ball.

We firstly define $l^2-$ball with radius $r$ in Euclidean space $\mathbb{R}^n$, it can be formally written as

$$\mathbb{B}(q, r) \triangleq \{x \mid \|x - q\|_2 \leq r\}.$$

Concretely, for any radius $r > 0$ define the $r-$neighborhood of $\Omega$ by

$$\Omega_r = \{x \in \mathcal{X} : \text{dist}(x, \Omega) \leq r\},$$

which is equivalently the Minkowski sum $\Omega + \mathbb{B}(0, r)$. By standard volume growth estimates (e.g., from Steiner's formula), the volume of $\Omega_r$ satisfies

$$Vol(\Omega_r) \leq Vol(\Omega) + \mathcal{O}(r),$$

for some constant $C_n > 0$ depending on the geometry of $\Omega$ and the ambient dimension $n$. Therefore, by choosing $r = \mathcal{O}(\epsilon^{1/n})$, we can ensure that $Vol(\Omega_r) \leq 2\epsilon$ for sufficiently small $\epsilon$. This allows us to recover a uniform dissipation on the complement $\mathcal{X} \setminus \Omega_r$.

To analyze the local behavior, if $x_t$ is in the small bad region $\Omega$. In such case, the extreme point can be write as follow:

$$
\begin{aligned}
\sup_t \mathcal{L}_f V(x_t) &\leq \mathcal{L}_f V(x_t') + \sup_t \|\mathcal{L}_f V(x_t') - \mathcal{L}_f V(x_t)\| \\
&\leq -\lambda V(x_t') + \sup_t \|\langle f(x_t', u_t'), \nabla V(x_t')\rangle - \langle f(x_t, u_t), \nabla V(x_t)\rangle\| \quad (31) \\
&\leq -\lambda V(x_t') + (\|\nabla f\|_\infty \|\nabla V\|_\infty + \|\nabla\nabla V\|\|f\|_\infty) \sup_t \|x_t' - x_t\|,
\end{aligned}
$$

where the derivation from this inequality is based on the regularity assumptions of $f$, $V$ and $u$. Here, the extreme point $x_t$ is lying in the bad region $\Omega$, and $x_t'$ is just outside of a covering region $\Omega_r$. Since $\Omega$ is always contained in a pre-defined region $\Omega_r$. We can always find $x_t'$, with the distance between $x_t$ and $x_t'$ bounded by $c_1 \epsilon^{\frac{1}{n}}$ with $c_1 > 0$. The fact can be easy to derive based on the geometric properties - the volume of hyper-sphere. According to this fact, we have

$$
\begin{aligned}
&- \lambda V(x_t') + (\|\nabla f\|_\infty \|\nabla V\|_\infty + \|\nabla\nabla V\|\|f\|_\infty) \sup_t \|x_t' - x_t\| \\
&\leq - \lambda V(x_t') + (\|\nabla f\|_\infty \|\nabla V\|_\infty + \|\nabla\nabla V\|\|f\|_\infty) c\epsilon^{\frac{1}{n}} \\
&\leq - (1 - \epsilon_1)\lambda V(x_t') - \epsilon_1 \lambda V(x_t') + (\|\nabla f\|_\infty \|\nabla V\|_\infty + \|\nabla\nabla V\|\|f\|_\infty) c\epsilon^{\frac{1}{n}} \quad (32) \\
&\leq - (1 - \epsilon_1)\lambda V(x_t') \underbrace{-\epsilon_1 \lambda V(x_t') + 2cB^2 \epsilon^{\frac{1}{n}}}_{\leq C\epsilon^{\frac{1}{n}}}.
\end{aligned}
$$

According to the radially unbounded of CLBF see Proposition B.4, we have the property that $V(x_t) \geq \mu\|x_t\|^2$ with some $\mu > 0$. Then, we have the following result such that

$$
-(1-\epsilon_1)\lambda V(x_t') - \epsilon_1 \lambda V(x_t') + 2cB^2 \epsilon^{\frac{1}{n}} \leq -(1-\epsilon_1)\lambda V(x_t'), \quad \forall x_t' \in \mathcal{X} \setminus \mathbb{B}(0, \sqrt{\frac{2cB^2 \epsilon^{\frac{1}{n}}}{\epsilon_1 \lambda \mu}}). \quad (33)
$$

**Step 2.** Probabilistic convergence via a supermartingale

**Stochastic model.** Assume the controlled dynamics via diffusion policy are given by

$$
\mathrm{d}x_t = f(x_t, u_t)\,\mathrm{d}t + \gamma_2\,\Sigma(x_t)\,\mathrm{d}W_t,
$$

where $W_t$ is an $n$-dimensional Wiener noise, Let $h > 0$ be a fixed time step and define the discrete-time sequence

$$
Z_k := V(x_{t_k}), \quad \text{where } t_k = kh.
$$

Let $\mathcal{F}_k = \sigma(x_0, x_1, \ldots, x_k)$ be the associated filtration.

**Itô increment decomposition.** Applying Itô's formula to $V(x_t)$ over interval $[t_k, t_{k+1}]$ yields:

$$
\begin{aligned}
Z_{k+1} - Z_k &= \int_{t_k}^{t_{k+1}} \mathcal{L}_f V(x_s, u_s)\,\mathrm{d}s + \frac{\gamma_2^2}{2} \int_{t_k}^{t_{k+1}} \mathrm{tr}\left(\Sigma^\top \nabla^2 V(x_s)\Sigma\right)\,\mathrm{d}s \\
&\quad + \gamma_2 M_k,
\end{aligned} \quad (34)
$$

where $M_k$ is a bounded martingale increment with $\mathbb{E}[M_k \mid \mathcal{F}_k] = 0$, and

$$
|M_k| \leq \Sigma_{\max}\|\nabla V\|_\infty \sqrt{h}.
$$

**Uniform dissipation outside small region.** Let $\widetilde{\mathcal{X}} := \mathcal{X} \setminus \mathbb{B}(0, r_\epsilon)$, where

$$
r_\epsilon = \sqrt{\frac{2cB^2 \epsilon^{1/n}}{\epsilon_1 \lambda \mu}}.
$$

From Step 1, we have

$$
\mathcal{L}_f V(x) \leq -(1 - \epsilon_1)\lambda V(x), \quad \forall x \in \widetilde{\mathcal{X}}.
$$

Moreover, if $\gamma_2$ is sufficiently small, then exist $\epsilon_2 > 0$ when the CLBF is radially unbounded (see Proposition B.4)

$$
\frac{\gamma_2^2}{2} \mathrm{tr}(\Sigma^\top \nabla^2 V(x)\Sigma) \leq \epsilon_2 \lambda V(x).
$$

Combining these, we obtain from (34):

$$\mathbb{E}[Z_{k+1} - Z_k \mid \mathcal{F}_k] \leq -\lambda_1 h Z_k, \quad \lambda_1 := (1 - \epsilon_1 - \epsilon_2)\,\lambda.$$

Thus, $\{Z_k\}$ is a nonnegative supermartingale.

**Concentration via Azuma–Hoeffding [31].** Since the trajectory generated by diffusion policy is always bounded, there exists a constant $\Delta_{\max} = \mathcal{O}(hB + \gamma_2\sqrt{hB})$ such that

$$|Z_{k+1} - Z_k| \leq \Delta_{\max}, \quad \text{a.s.}$$

By Azuma–Hoeffding inequality, for any $N \in \mathbb{N}$ and $\eta > 0$:

$$p\left(Z_N - Z_0 + \lambda_1 \sum_{k=0}^{N-1} h Z_k \geq \eta\right) \leq \exp\left(-\frac{2\eta^2}{N\Delta_{\max}^2}\right).$$

Choosing $\eta = \Delta_{\max}\sqrt{N \log N}$, we ensure the sum is finite as $\sum_{N=1}^{\infty} p(Z_N - Z_0 + \lambda_1 \sum_{k=0}^{N-1} Z_k) \leq \sum_{N=1}^{\infty} N^{-2} < \infty$. Then, by the first Borel–Cantelli lemma:

$$p\left(Z_{k+1} - Z_k + \lambda_1 h Z_k \geq 0 \text{ infinitely often}\right) = 0,$$

which implies that almost surely:

$$Z_{k+1} - Z_k \leq -\lambda_1 h Z_k \quad \text{for all large } k.$$

Therefore,

$$Z_k \leq (1 - \lambda_1 h)^k Z_0 \quad \text{a.s.}$$

This implies that once the initial state $x_0$ lies within the safe set, safety is almost surely guaranteed, as the CLBF exhibits an almost monotonic decrease.

**Return to continuous time and add influence of** $\Omega$. Replacing the discrete time $k$ to continuous time $t/h$ and incorporating the worst-case additive term from the bad region $\Omega$, which is bounded by $C\epsilon^{1/n}$ in Equation (32), we invoke the Comparison Lemma C.5 to conclude:

$$V(x_t) \leq \exp(-\lambda_1 t)V(x_0) + \frac{C\epsilon^{1/n}}{\lambda_1}(1 - \exp(-\lambda_1 t)), \quad \forall t \geq 0, \quad \text{a.s.}$$

where $0 < \lambda_1 < \lambda$. This completes the probabilistic part of the proof.

$\square$

# D More Details About Experiments

## D.1 Tasks

**Inverted Pendulum.** The state of the inverted pendulum is given by $x = [\theta, \dot{\theta}]$, where $\theta$ is the angular displacement from the upright position (measured counterclockwise from the vertical) and $\dot{\theta}$ is the angular velocity. The control input is a torque $u$ applied at the pivot. The system is parameterized by the pendulum mass $m$, length $l$, and moment of inertia $I = ml^2$. The dynamics are given by the second-order nonlinear equation given by

$$\dot{x} = \begin{bmatrix} \dot{\theta} \\ -\frac{mgl}{I}\sin\theta + \frac{1}{I}u \end{bmatrix},$$

and $g$ is the gravitational acceleration. We consider a stabilization task at the upright equilibrium. The goal set is $x_* = 0$, the safe set is defined as $\mathcal{X}_{\text{safe}} = \{x : |\theta| \le 0.5\,\text{rad} \wedge \|x\| \le 2\}$, and the unsafe set is defined as $\mathcal{X}_{\text{unsafe}} = \{x : |\theta| \ge \pi/2 \vee \|x\| \ge 2.5\}$.

**Car (Kin).** The kinematic single-track car model is to catch a reference path. The reference path is parameterized by its linear velocity $v_{\text{ref}}$, acceleration $a_{\text{ref}}$, heading $\psi_{\text{ref}}$, and angular velocity $\omega_{\text{ref}}$.

The state of the path-centric model is defined as $x = [x_e, y_e, \delta, v_e, \psi_e]$, where $(x_e, y_e)$ denotes the position error in the Frenet frame, $\delta$ is the steering angle, $v_e$ is the velocity error and $\psi_e$ is the heading error. The control input is $u = [v_\delta, a_{\text{long}}]$, representing the steering rate and the longitudinal acceleration, respectively. The dynamics takes the form $\dot{x} = f(x) + g(x)u$, where

$$f(x) = \begin{bmatrix} v\cos(\psi_e) - v_{\text{ref}} + \omega_{\text{ref}}y_e \\ v\sin(\psi_e) - \omega_{\text{ref}}x_e \\ 0 \\ -a_{\text{ref}} \\ \frac{v}{l_r+l_f}\tan(\delta) - \omega_{\text{ref}} \end{bmatrix}, \quad g(x) = \begin{bmatrix} 0 & 0 \\ 0 & 0 \\ 1 & 0 \\ 0 & 1 \\ 0 & 0 \end{bmatrix},$$

with $v = v_e + v_{\text{ref}}$, and $l_f$, $l_r$ denoting the distances from the vehicle's center of mass to the front and rear axles, respectively. The goal state is defined as $x_* = 0$. Since the reference heading and position do not explicitly appear in the dynamics, this formulation describes a tracking task rather than a reach-avoid task.

**Car (Slip).** The dynamic single-track (sideslip) car model is used for a trajectory tracking task, where the objective is to follow a reference path. This model captures more complex vehicle behavior than the kinematic model by incorporating lateral dynamics and tire slip, which become significant at higher speeds or during aggressive maneuvers. The state of the model is $x = [x_e, y_e, \delta, v_e, \psi_e, \dot{\psi}_e, \beta]$, where $x_e, y_e$ are lateral and longitudinal errors, $\delta$ is the steering angle, $v_e$ is the velocity error, $\psi_e$ is the heading error, $\dot{\psi}_e$ is its time derivative, and $\beta$ is the sideslip angle. The control inputs $u = [v_\delta, a_{\text{long}}]$ are the same as in the kinematic model. The dynamics takes the control form $\dot{x} = f(x) + g(x)u$, where

$$f(x) = \begin{bmatrix} v\cos(\psi_e) - v_{\text{ref}} + \omega_{\text{ref}}y_e \\ v\sin(\psi_e) - \omega_{\text{ref}}x_e \\ 0 \\ 0 \\ \dot{\psi}_e \\ -\frac{\mu m}{vI_z(l_r+l_f)}(l_f^2 C_{Sf}gl_r + l_r^2 C_{Sr}gl_f)(\dot{\psi}_e + \omega_{\text{ref}}) + \frac{\mu m}{I_z(l_r+l_f)}(l_r C_{Sr}gl_f - l_f C_{Sf}gl_r)\beta + \frac{\mu m}{I_z(l_r+l_f)}(l_f C_{Sf}gl_r)\delta \\ \left(\frac{\mu}{v^2(l_r+l_f)}(C_{Sr}gl_fl_r - C_{Sf}gl_rl_f) - 1\right)(\dot{\psi}_e - \omega_{\text{ref}}) - \frac{\mu}{v(l_r+l_f)}(C_{Sr}gl_f C_{Sf}gl_r)\beta + \frac{\mu}{v(l_r+l_f)}(C_{Sf}gl_r)\delta \end{bmatrix},$$

$$g(x) = \begin{bmatrix} 0 & 0 \\ 0 & 0 \\ 1 & 0 \\ 0 & 1 \\ 0 & 0 \\ 0 & 0 \\ 0 & 0 \end{bmatrix},$$

with $v = v_e + v_{\text{ref}}$, and parameters $l_f, l_r, C_{Sf}, C_{Sr}, \mu$. $g$ denotes gravitational acceleration. The goal state is defined as $x_* = 0$.

**Segway.** We consider the Segway obstacle avoidance task, where the system must move forward while avoiding a circular obstacle. Successful avoidance requires the Segway to temporarily tilt forward to maneuver around the obstruction. The state of the system is defined as $x = [p, \theta, v, \omega]$, where $p$ is the horizontal position, $\theta$ is the tilt angle and $v$, $\omega$ are the corresponding linear and angular velocities. The control input $u$ is a horizontal force applied at the base. We assume the wheel's vertical position remains at zero and the Segway length is normalized to one. The obstacle is modeled as a circle of radius $0.1$ centered at $(0, 1)$. The position of the Segway's top is given by $(p_x, p_y) = (p + \sin\theta, \cos\theta)$. The unsafe set is defined as

$$\mathcal{X}_{\text{unsafe}} = \left\{ x \mid \sqrt{p_x^2 + (p_y - 1)^2} \leq 0.1 \right\},$$

and the safe set is

$$\mathcal{X}_{\text{safe}} = \left\{ x \mid \sqrt{p_x^2 + (p_y - 1)^2} \geq 0.15 \right\}.$$

Let $M$ be the base mass, $m$ and $J$ the mass and moment of inertia of the body, and $l$ the distance from the base to the center of mass. Define total mass $M_t = M + m$ and total inertia $J_t = J + ml^2$. The gravitational constant is denoted $g$. The dynamics are expressed in form as $\dot{x} = f(x) + g(x)u$, where

$$f(x) = \begin{bmatrix} v \\ \omega \\ \frac{g\sin\theta\cos\theta + \lambda_1 v\cos\theta + \lambda_2 v - l\omega^2\sin\theta}{\cos\theta - \frac{M_t J_t}{m^2 l^2} + \lambda_9} \\ \frac{\lambda_3 v\cos\theta + \lambda_4 v - \frac{M_t g}{ml}\sin\theta - \omega^2\sin\theta\cos\theta}{\cos^2\theta - \frac{M_t J_t}{m^2 l^2} + \lambda_9} \end{bmatrix}, \quad g(x) = \begin{bmatrix} 0 \\ 0 \\ \frac{\frac{\lambda_6}{M_t}(\lambda_5 + \cos\theta)}{\cos^2\theta - \frac{M_t J_t}{m^2 l^2} + \lambda_9} \\ \frac{\frac{\lambda_8 l}{J_t}(\cos\theta + \lambda_7)}{\cos^2\theta - \frac{M_t J_t}{m^2 l^2} + \lambda_9} \end{bmatrix}.$$

Here, $\lambda_i$ denote intermediate constants derived from model parameters and simplifications. This formulation captures the coupled nonlinear dynamics and their dependence on tilt for obstacle avoidance.

**Neural Lander.** The state of the Neural Lander model is defined as $x = [p_x, p_y, p_z, v_x, v_y, v_z]$, where $[p_x, p_y, p_z]$ denotes position and $[v_x, v_y, v_z]$ velocity. The control input is $u = [f_x, f_y, f_z]$, with $p_z$ defined to be positive in the upward direction. This model is parameterized by the mass $m$ and system dynamics are expressed as $\dot{x} = f(x) + g(x)u$, where

$$f(x) = \begin{bmatrix} v_x \\ v_y \\ v_z \\ F_{a1}/m \\ F_{a2}/m \\ F_{a3}/m - g \end{bmatrix}, \quad g(x) = \begin{bmatrix} 0 & 0 & 0 \\ 0 & 0 & 0 \\ 0 & 0 & 0 \\ 1/m & 0 & 0 \\ 0 & 1/m & 0 \\ 0 & 0 & 1/m \end{bmatrix},$$

and $g$ is the gravitational acceleration. The term $F_a = [F_{a1}, F_{a2}, F_{a3}]$ represents the aerodynamic disturbance due to the ground effect. The goal state is defined as $x_* = 0$, the safe set as $\mathcal{X}_{\text{safe}} = \{x : p_z \geq -0.05 \wedge \|x\| \leq 3\}$, and the unsafe set as $\mathcal{X}_{\text{unsafe}} = \{x : p_z \leq -0.3 \vee \|x\| \geq 3.5\}$.

**2D Quadrotor.** The state vector of the 2D quadrotor model is defined as $x = [p_x, p_z, \theta, v_x, v_z, \dot{\theta}]$, where $[p_x, p_z]$ denotes position, $\theta$ is the pitch angle, and $[v_x, v_z]$, $\dot{\theta}$ represent the corresponding velocities. The control input is $u = [u_1, u_2]$, with $p_z$, $u_1$, and $u_2$ taken to be positive in the upward direction. The system is characterized by the mass $m$, moment of inertia $I$, and rotor arm length $r$. The dynamics are described by the system $\dot{x} = f(x) + g(x)u$, where

$$f(x) = \begin{bmatrix} v_x \\ v_z \\ \dot{\theta} \\ 0 \\ -g \\ 0 \end{bmatrix}, \quad g(x) = \begin{bmatrix} 0 & 0 \\ 0 & 0 \\ 0 & 0 \\ \frac{1}{m}\sin\theta & \frac{1}{m}\sin\theta \\ \frac{1}{m}\cos\theta & \frac{1}{m}\cos\theta \\ \frac{r}{I} & -\frac{r}{I} \end{bmatrix},$$

and $g$ denotes the gravitational constant. The unsafe set $\mathcal{X}_{\text{unsafe}}$ is defined as the region occupied by obstacles, while the safe set $\mathcal{X}_{\text{safe}}$ is specified as a 0.1 m offset from the obstacle boundaries.

**3D Quadrotor.** The state of the 9-dimensional quadrotor model is defined as $x = [p_x, p_y, p_z, v_x, v_y, v_z, \phi, \theta, \psi]$, where $p_i$ and $v_i$ denote positions and velocities, and $\phi, \theta, \psi$ are the roll, pitch, and yaw angles, respectively. The control input is given by $u = [f, \dot{\phi}, \dot{\theta}, \dot{\psi}]$. This model is parameterized by the mass $m$, and system dynamics are expressed as a form, $\dot{x} = f(x) + g(x)u$, with

$$
f(x) = \begin{bmatrix} v_x \\ v_y \\ v_z \\ 0 \\ 0 \\ -g \\ 0 \\ 0 \\ 0 \end{bmatrix}, \quad g(x) = \begin{bmatrix} 0 & 0 & 0 & 0 \\ 0 & 0 & 0 & 0 \\ 0 & 0 & 0 & 0 \\ -\frac{1}{m}\sin\theta & 0 & 0 & 0 \\ \frac{1}{m}\cos\theta\sin\phi & 0 & 0 & 0 \\ \frac{1}{m}\cos\theta\cos\phi & 0 & 0 & 0 \\ 0 & 1 & 0 & 0 \\ 0 & 0 & 1 & 0 \\ 0 & 0 & 0 & 1 \end{bmatrix},
$$

where $g$ denotes the gravitational acceleration. The goal state is defined as $x_* = \{0\}$, the safe set as $\mathcal{X}_{\text{safe}} = \{x : p_z \geq 0 \land \|x\| \leq 3\}$, and the unsafe set as $\mathcal{X}_{\text{unsafe}} = \{x : p_z \leq -0.3 \lor \|x\| \geq 3.5\}$.

**F-16.** The F-16 aircraft is modeled as a nonlinear dynamical system with a 16-dimensional state vector $x \in \mathbb{R}^{16}$ and a 4-dimensional control input vector $u \in \mathbb{R}^4$. Each component of the state vector has a physical interpretation as follows:

$$
x = [V_T, \alpha, \beta, \phi, \theta, \psi, P, Q, R, p_n, p_e, h, \text{pow}, \text{int}_{N_z}, \text{int}_{ps}, \text{int}_{Ny_r}]
$$

- $V_T$: True airspeed (ft/s) — the speed of the aircraft relative to the surrounding air.
- $\alpha$: Angle of attack (rad) — the angle between the aircraft's longitudinal axis and the oncoming airflow.
- $\beta$: Sideslip angle (rad) — the angle between the aircraft's heading and its actual flight path laterally.
- $\phi$: Roll angle (rad) — the rotation of the aircraft about its longitudinal axis.
- $\theta$: Pitch angle (rad) — the rotation of the aircraft about its lateral axis.
- $\psi$: Yaw angle (rad) — the rotation about the vertical axis, determining heading.
- $P$: Roll rate (rad/s) — angular velocity about the longitudinal (roll) axis.
- $Q$: Pitch rate (rad/s) — angular velocity about the lateral (pitch) axis.
- $R$: Yaw rate (rad/s) — angular velocity about the vertical (yaw) axis.
- $p_n$: Northward position (ft) — horizontal displacement in the north direction.
- $p_e$: Eastward position (ft) — horizontal displacement in the east direction.
- $h$: Altitude (ft) — vertical position above sea level.
- pow: Engine power state — internal dynamic state representing engine thrust lag.
- $\text{int}_{N_z}$: Integrator state for vertical acceleration (normal G-force) regulation.
- $\text{int}_{ps}$: Integrator state for stability-axis roll rate tracking.
- $\text{int}_{Ny_r}$: Integrator state for lateral acceleration and yaw rate tracking.

The control input vector is:
$$
u = [N_z^{\text{ref}}, p_s^{\text{ref}}, Ny_r^{\text{ref}}, \text{throttle}]
$$

- $N_z^{\text{ref}}$: Reference normal acceleration (G-forces) — controls pitch via the elevator.
- $p_s^{\text{ref}}$: Reference roll rate — governs rolling motion via ailerons and rudder.
- $Ny_r^{\text{ref}}$: Reference combination of side acceleration and yaw rate — maintains coordinated flight and yaw control.

- `throttle`: Throttle command — scalar value in [0, 1], where 0 represents idle and 1 represents full throttle.

This system is **not affine in control**; that is, it cannot be expressed as $\dot{x} = f(x) + g(x)u$ because the inputs pass through a nonlinear autopilot that internally computes actuator commands, which are then applied via a complex, nonlinear aircraft model involving aerodynamic forces, actuator dynamics, and engine lag.

**Goal Set:**
$$x_\star = \left\{ x \in \mathbb{R}^{16} \mid \theta + \alpha \geq 0,\ |\phi| \leq 0.1,\ |P| \leq 0.25,\ h \geq 1000 \right\}$$

**Safe Set:**
$$\mathcal{X}_{\text{safe}} = \left\{ x \in \mathbb{R}^{16} \mid h \geq 500,\ 0.95 \cdot x_{\min} \leq x \leq 0.95 \cdot x_{\max},\ x \notin \mathcal{B}_{\text{goal}} \right\}$$

where $\mathcal{B}_{\text{goal}}$ is a buffer region around the goal set defined as:
$$\mathcal{B}_{\text{goal}} = \left\{ x \in \mathbb{R}^{16} \mid \theta + \alpha \geq -0.2,\ |\phi| \leq 0.2,\ |P| \leq 0.5,\ h \geq 800 \right\}$$

**Unsafe Set:**
$$\mathcal{X}_{\text{unsafe}} = \left\{ x \in \mathbb{R}^{16} \mid h \leq 100 \right\}$$

# E   Implementation

---

**Algorithm 1:** $S^2$Diff

---

**Input:** Distribution of initial states $\mathcal{D}_{x_0}$, model dynamics $f$, nominal policy, number of training epochs $K$

**Output:** Certificate function (CLBF) $V_K$

Initialize the certificate function (CLBF) $V_0$ with parameters $\theta$;
**for** *epoch $k = 1$ to $K$* **do**

    // === Phase 1:  Guided Trajectory Sampling ===
    Initialize an empty dataset of new trajectories $\mathcal{D}$;
    **for** *each initial state $x_0$ in a batch sampled from $\mathcal{D}_{x_0}$* **do**

        Generate one full trajectory via a guided denoising process by maximizing Equations (6) and (8);
        Sample a clean trajectory $U^0$ by applying the reverse diffusion process (Equation (10)) with model dynamics $f$, starting from noise;
        The process is guided at each step by the current CLBF $V_{k-1}$;
        Add the resulting trajectory $U^0$ to the dataset $\mathcal{D}$;

    // === Phase 2:  CLBF Update ===
    Use the entire newly generated dataset $\mathcal{D}$ for training;
    Update CLBF parameters by performing gradient descent on the loss from Equation (11), using trajectories from $\mathcal{D}$, obtain $V_k$;

---

In Phase 1, we use the current CLBF $V_{k-1}$ to guide a model-based diffusion [32] sampler to generate a batch of trajectories. This is done by sampling from a CLBF-shaped target distribution $p(U)$, defined in Equation (6) and (8), and using the reverse diffusion process. At each denoising step, we approximate the score $\nabla \log p(U^i)$ via sequential Monte Carlo, leveraging the known dynamics model $f(x, u)$.

Importantly, our diffusion process is not learned via neural network training. Instead, it is a fully algorithmic, model-based diffusion guided by the CLBF through the structure of the target distribution $p(U)$. This distribution is explicitly designed to assign higher likelihood to trajectories that satisfy the safety and stability criteria, i.e., $p_{\text{safe}}$ and $p_{\text{stable}}$. As a result, the diffusion process is biased toward generating CLBF-compliant trajectories without requiring gradient-based training of a score network. While there is no learned diffusion model, the objective $p(U)$ still plays a central role—it guides the sampling, not learning.

In Phase 2, the CLBF is updated using the sampled trajectories by minimizing the supervised loss defined in Equation (11). This alternating process is repeated over training epochs, allowing the CLBF and the sampler to iteratively improve.

# F  More Experiment Results

We also present additional experimental results, including extended ablation studies and further quantitative analysis. Note that all inference time evaluations were conducted on the same device: Intel i9-13900 CPU with one RTX 4090 GPU.

## F.1  More Ablation Studies

**Ablation in Trajectory Length** We conduct an ablation study to evaluate the impact of trajectory length on the performance of $S^2$Diff in neural lander, as shown in Table 6. Short trajectories (length 2) lead to low evaluation time but suffer from poor safety (35%) due to limited foresight. Increasing the trajectory length to 5 significantly improves both safety (100%) and accuracy, achieving the best performance in terms of tracking error. However, further increasing the trajectory length results in diminishing returns for safety and degraded tracking performance, along with substantially higher evaluation time. This suggests that moderate trajectory lengths strike a good balance between efficiency, safety, and control accuracy, while excessively long rollouts introduce variance and unnecessary computational cost.

Table 6: Control performance of $S^2$Diff under different trajectory lengths in neural lander.

| Trajectory length | Eval. time (ms) | Safety rate | $\|x - x_\star\|$ |
|---|---|---|---|
| 2 | $\mathbf{28.9 \pm 0.4}$ | 35% | $0.08 \pm 0.05$ |
| 5 | $35.4 \pm 0.7$ | 100% | $\mathbf{0.06 \pm 0.02}$ |
| 10 | $105.1 \pm 2.4$ | 100% | $0.09 \pm 0.06$ |
| 15 | $126.3 \pm 1.5$ | 100% | $0.17 \pm 0.09$ |
| 20 | $237.5 \pm 6.8$ | 100% | $0.26 \pm 0.07$ |
| 50 | $598.6 \pm 25.0$ | 100% | $0.35 \pm 0.17$ |

**Parameterization of $V$.** We observed that the quadratic-form parameterization $V(x) = w(x)^\top w(x)$, as used in [13], can result in a high failure probability for systems with non-convex constraints, such as the 2D quadrotor, Segway, and F-16. In contrast, employing a general neural network parameterization for $V$ improves both training stability and control performance.

## F.2 Other Quantitative Results

### F.2.1 Inverted Pendulum

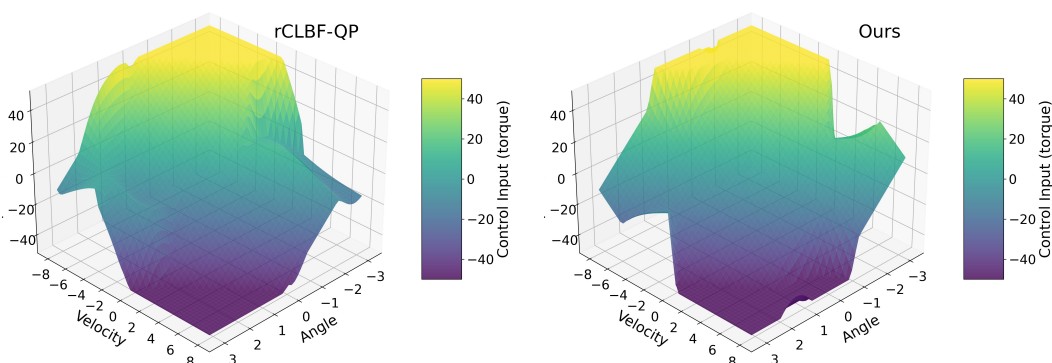

Figure 6: Comparison of policy landscapes for the inverted pendulum task — rCLBF-QP (left) vs. ours (right). Each surface visualizes the controller's output torque over a 2D slice of the state space (pendulum angle vs. angular velocity). The surface height and color both represent the magnitude of the control input, providing a physical interpretation of how each policy maps states to actions. Our method yields a smooth and symmetric policy around the upright equilibrium (angle $= 0$, velocity $= 0$), which is desirable for stability and generalization. In contrast, the rCLBF-QP policy produces irregular and less structured patterns, likely due to its step-wise greedy updates and slack variable usage.

### F.2.2 Car Tracking

Figure 7 and Figure 8 compare the control performance of rCLBF-QP and $S^2$Diff in car tracking tasks. While both methods closely follow the reference trajectory, $S^2$Diff achieves a more globally consistent performance with notably lower tracking error across time. The error profile of $S^2$Diff (Figure 8, right) shows reduced oscillation and slower error accumulation compared to rCLBF-QP (Figure 7, right), indicating improved stability and accuracy in long-horizon tracking.

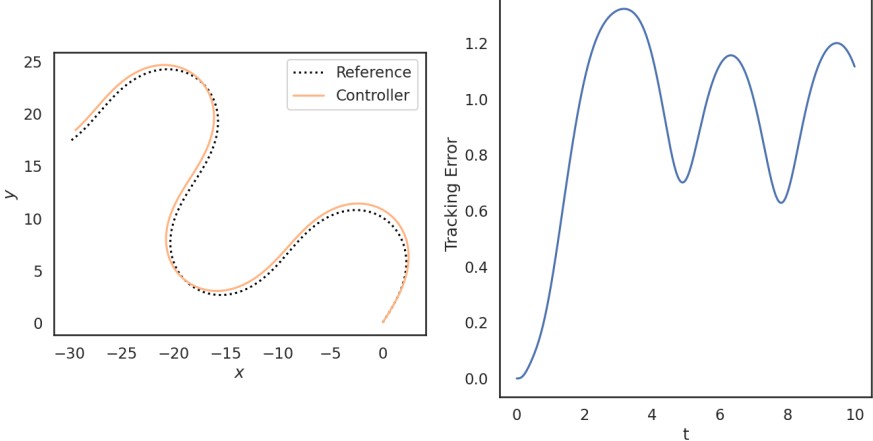

Figure 7: Control performance of rCLBF-QP in car tracking tasks. The left plot shows the reference and controlled trajectories, while the right plot depicts the tracking error over time.

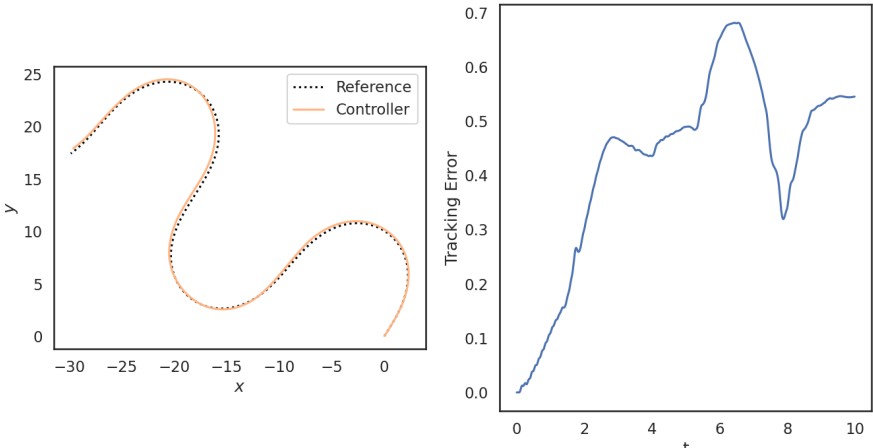

Figure 8: Control performance of $S^2$Diff in car tracking tasks. The left plot shows the reference and controlled trajectories, while the right plot depicts the tracking error over time.

### F.2.3 Segway

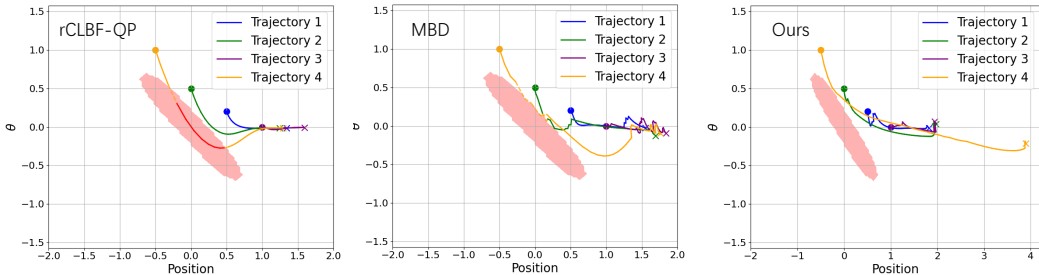

Figure 9: Comparison of control performance across three methods — rCLBF-QP (left), MBD (center), and Ours (right). The objective of this task is to stabilize the system to the target point $(2, 0)$. The figure shows that rCLBF-QP struggles to ensure stability due to globally inconsistent behavior induced by its step-wise QP formulation. MBD approaches the boundary of the unsafe set and exhibits unstable performance with higher variance. In contrast, our method achieves safe and consistent control, with most initial conditions stabilizing close to the target point $(2, 0)$. This demonstrates that our algorithm ensures more globally consistent behavior and delivers more stable performance than MBD, guided by the learned CLBF.

### F.2.4    Quadrotor

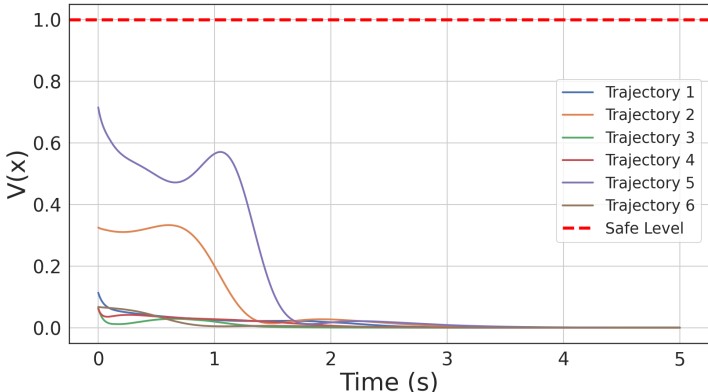

Figure 10: Change in the scalar value of the CLBF under the control policy of $S^2$Diff for the 2D quadrotor. The color of each trajectory corresponds to Figure 4. The scalar values of the CLBF from different initial conditions under $S^2$Diff exhibit a nearly monotonic decrease, closely aligning with Lyapunov theory.

### F.2.5    Neural Lander

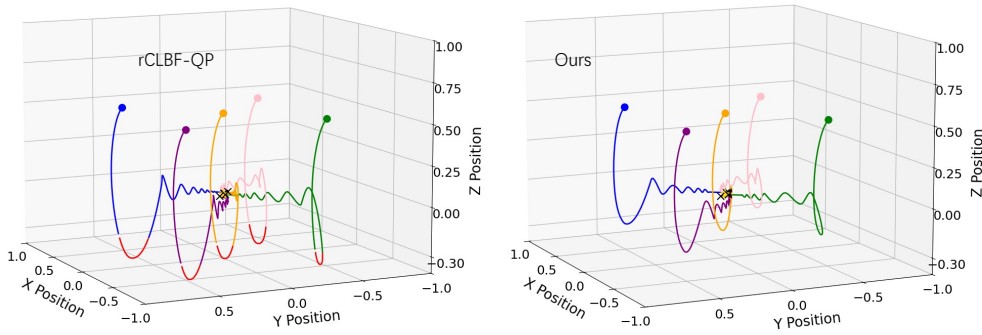

Figure 11: Comparison of 3D trajectories under rCLBF-QP (left) and our method (right). Each colored trajectory represents a system evolution from a different initial condition, with the black star indicating the target. The rCLBF-QP controller fails to guarantee global safety when the slack variable is reduced, leading to constraint violations and unsafe behavior. In contrast, our method consistently maintains safe and stable trajectories across all initial conditions.

### F.2.6 F-16

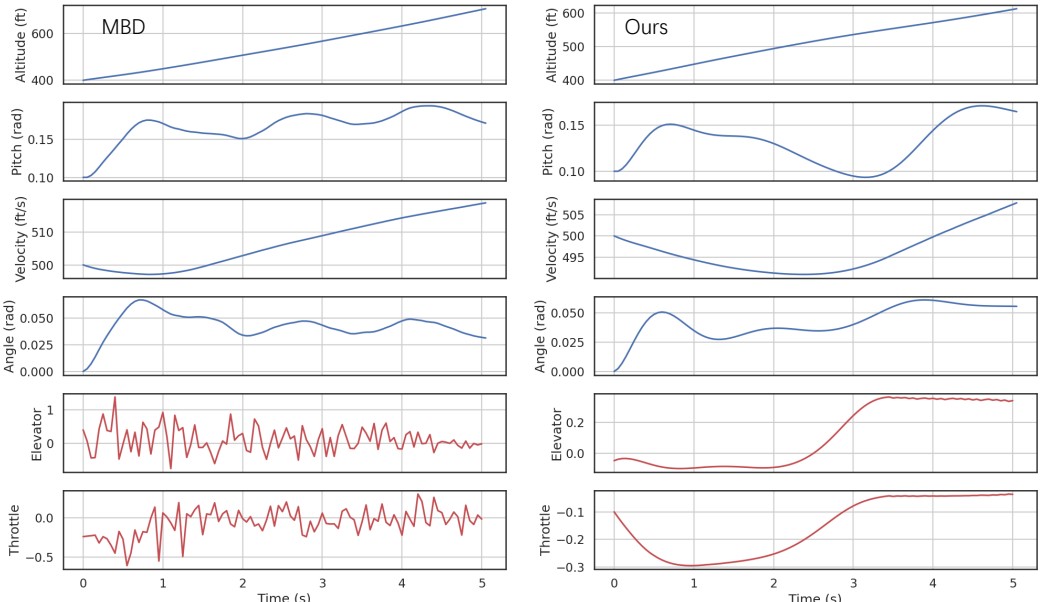

Figure 12: Comparison of F-16 control performance between MBD (left) and our method (right). The plots illustrate the evolution of altitude, pitch angle, velocity, angle of attack, and control inputs (elevator and throttle) over time. Our method achieves smoother and more stable flight dynamics, with significantly less oscillation in pitch and angle of attack, and well-regulated velocity recovery. Control inputs remain continuous and low-frequency, indicating efficient and robust control. In contrast, MBD shows high-frequency fluctuations in both elevator and throttle signals, suggesting aggressive or unstable control behavior.

# G  Implementation of Baselines.

**rCLBF-QP.**  The implementation of this baseline, including network architectures and hyperparameters, strictly follows the settings described in the original paper [13]. The corresponding code is publicly available at `https://github.com/MIT-REALM/neural_clbf/tree/main`.

**MPC.**  This baseline employs model predictive control (MPC), solved using the Gurobi optimizer at each time step. The implementation strictly follows the settings described in [13], including a time discretization of $0.1$ seconds and a lookahead horizon of 5 steps. The MPC is configured to track reference trajectories while satisfying both dynamic and control constraints.

**MBD.**  This baseline is implemented following the methodology and hyperparameter settings outlined in the original paper [33]. To ensure a fair comparison with MPC, the MBD controller uses a time step of $0.1$ seconds and a planning horizon of 5 steps. The controller is trained to generate control sequences that track reference trajectories while satisfying learned dynamics and safety constraints. The implementation code is available at `https://github.com/LeCAR-Lab/model-based-diffusion`.

**Ours.**  The following Table 7 provides the experimental details, including the network architecture and other relevant hyperparameters.

Table 7: Hyperparameter configurations for eight tasks using $S^2$Diff.

| Task | NN Architecture[†] | Safe Level $c$ | Temp. | Loss Coeff.[‡] | Traj. Len. |
|------|------|------|------|------|------|
| Inverted Pendulum | 3×64 | 1 | 0.1 | 1,1 | 5 |
| Car (Kin.) | 3×64 | 1 | 0.1 | 1,1 | 5 |
| Car (Slip) | 3×64 | 1 | 0.1 | 1,1 | 5 |
| Segway | 3×64 | 1 | 0.1 | 1,1 | 5 |
| Neural Lander | 3×64 | 10 | 0.1 | 1,1 | 5 |
| 2D Quad | 3×64 | 1 | 0.1 | 1,1 | 5 |
| 3D Quad | 3×64 | 10 | 0.1 | 1,1 | 5 |
| F-16 | 3×128 | 10 | 0.1 | 1,1 | 5 |

[†] Format "L×H" denotes L layers with H hidden units per layer.
[‡] Two coefficients $(\alpha_1, \alpha_2)$ represent weights for Lie derivative and discrete differences and CLBF losses, respectively.

