# OpenReview forum: "Safe and Stable Control via Lyapunov-Guided Diffusion Models"
_NeurIPS.cc/2025/Conference — NeurIPS 2025 poster_

### Official Review · Reviewer_xnzn · 2025-06-11

**Clarity:** 2
**Significance:** 4
**Originality:** 3
**Rating:** 5
**Confidence:** 3

**Summary:**

This paper proposes the $S^2$ framework, which combines diffusion policies with Control Lyapunov Barrier Functions (CLBFs) to ensure safety and stability in control tasks. For this purpose, the authors propose an iterative training procedure that alternates between training a diffusion policy and a CLBF. The diffusion policy is trained to generate actions that are safe and stable, while the CLBF is trained to ensure that it captures accurately the safety and stability constraints of the system. The paper demonstrates the effectiveness of the $S^2$ framework through experiments on various control tasks (e.g., Inverted Pendulum, Car, Segway, Neural Lander, and F-16) and against various baselines, including robust CLBF-QP, MPC, Model-Based Diffusion, showing that it outperforms existing methods in terms of stability/convergence behavior and safety at a greater computational efficiency than MPC and Model-Based Diffusion.
An additional advantage of the proposed method compared to MPC and rCLBF-QP is that it also admits to non-control-affine systems, as demonstrated in the experiments on the F-16 aircraft.
Finally, the paper connects the proposed framework to Almost Lyapunov theory, showing that the policy almost-always guarantees convergence to the desired equilibrium point with exponential convergence rate.

**Questions:**

- How is the diffusion policy trained to ensure adherence to the CLBF constraints? Furthermore, how many iterations alternating between training the diffusion policy and the CLBF are performed? Is there a stopping criterion for this iterative training procedure? Finally, the reviewer would like to see an ablation study that compares the iterative training procedure with a joint training procedure of the diffusion policy and the CLBF. How do you sample the trajectories, which you mention in Section 3.2? What are the initial conditions?
- Could you plot the results of Table 3, preferably with a smaller discretization, in a Pareto fron style plot, to better visualize the trade-off between safety and stability/convergence? Furthermore, why is the best convergence rate achieved for an intermediate temperature value? The reviewer would expect the convergence characteristics to improve with a lower temperature. Could this possibly point to issues in the evaluation procedure and/or the definition of the evaluation metric for convergence?
- Re Table 4: Why does adding a term based on the discrete approximation of the Lie derivative actually improve the performance compared to the sole reliance on the analytical Lie derivative? This is unintuitive. Do the authors have an explanation for this?
- Re Results Tables and Appendix F: The selection of the discretization and the horizon for MPC seems arbitrary and might partly explain why the MPC is not able to accomplish high/perfect safety rates. Therefore, the reviewer would like to see different MPC configurations reported in the results tables (e.g., MPCs with short, medium and long horizons). One would expect the short-horizon MPC to be computationally more efficient but to achieve lower safety rates, while the long-horizon MPC to be computationally less efficient but to achieve higher safety rates.

The questions touch on the presentation and the quantitative evaluation in the paper. The reviewer expects that a satisfactory answer to these questions will be provided in the rebuttal and that the paper score could be adjusted accordingly.

**Ethical Concerns:**

["NO or VERY MINOR ethics concerns only"]

**Final Justification:**

The reviewer very much appreciates the author's responsiveness and the extensive explanations during the rebuttal and discussion phase. The reviewer values the contributions of the paper, the added ablations and baselines, and, most importantly, the changes that the authors laid out and promised for the final version of the paper.

Still, the reviewer needs to conclude that the scheduled changes are relatively substantive in quantity and that the final review justification cannot be _solely_ based on promised changes to the manuscript, but instead with a substantial share on the initially submitted manuscript.

Recognizing the author's effort and proposed improvements to the manuscript, the reviewer decides to elevate their score to a "5: Accept:", although the most appropriate and fair score would probably be a "4.5".

**Limitations:**

The paper mentions a single limitation at the end of the conclusion regarding its slower inference speed compared to QP methods. However, the reviewer imagines that there must be other limitations to this method and the authors should transparently discuss those in the main text and possibly more extensively in the appendix.

- For example, another limitation is that the proposed method does not formally guarantee convergence in all situations, but rather "just" almost-always guarantees.
- Related to the previous point, the reviewer understands that the "almost lypunov" guarantees only hold if the loss function actually converges to zero. Is this indeed the case? Please clarify this in the main text - preferably also in the introduction and the conclusion.
- In the introduction, you claim that "However, designing valid CBFs priors remains a challenge.". Indeed, while the proposed approach learns the CBFs, it still requires a manual definition of the safe state set, which should be explicitly discussed in the main text.
- Finally, another limitation is that the proposed method requires knowledge about the system dynamics, which is not always (accurately) available in practice.

**Paper Formatting Concerns:**

The paper currently does not contain page numbers. Also, the reviewer would prefer a ascending citation order, but this is a matter of personal preference and does not affect the review. Other than that, the reviwer does not have any formatting concerns.

**Quality:**

3

**Strengths And Weaknesses:**

## Strengths

- The paper is a timely contribution to the field of safe and stable control by integrating safety and stability guarantees into state-of-the-art diffusion policies.
- The proposed $S^2$ framework is an interesting, novel, and valuable contribution and offers interesting theoretical and technical insights into the integration of diffusion models with control theory.
- The proposed framework is comprehensively quantitatvely evaluated on a variety of control tasks and against various relevant baselines, demonstrating its effectiveness in terms of safety, stability, and computational efficiency.

## Weaknesses

- The current manuscript fails to clearly and in detail define the overall training procedure that consists of this iterative training procedure alternating between training the diffusion policy and the CLBF. For example, it is not clear according to which objectives the diffusion policy is trained in order to ensure adherence to the CLBF constraints. The reviewer recommends a clear and easily understandable description of the overall training procedure and the algorithm in the main text, before each of the components are described in detail. Also, the manuscript is missing illustrations and figures that illustrate the methodology and the training procedure.
- The quantitative evaluation is lacking in some aspects: (1) unconstrained diffusion policies should be included as a baseline, (2) different configurations of MPC should be included as a baseline (i.e, some with short, medium, and long horizons) in order to demonstrate that long-horizon MPC can achieve high safety rates, (3) adopting SafeDiffuser as a baseline would be interesting, (4) arguably **most importantly**, an evaluation metric for the task performance is missing: while the paper evaluates the convergence/stability of the methods, the safety rate, and the computational efficiency, it does not evaluate the task performance of the methods (i.e., reward, cost to goal, etc.).
- The paper shows the connections to Almost Lyapunov theory. However, the paper does not take the next step of actually applying Theorem 3.1 to practical examples. For the considered tasks, what is the epsilon value? Does this epsilon value together with the learned Lyapunov function satisfy the conditions/asssumptions? Can the authors conduct a post-training analysis of the learned CLBF function and how it connects to the Almost Lyapunov theory? This would be a valuable addition to the paper.

### Detailed Comments on Minor Issues

- Figure 1: The authors introduce a color map from "violation to satisfaction". However, it is using the same colormap as for the "update Lyapunov function". This is confusing and unclear. Please revise.
- Page 2, line 85: "... the stability properties of such methods are still unexplored". The reviewer disagrees with this statement as CBF+CLF approaches guarantee simultatenously safety and stability. Please clarify this statement.
- Section 2 (Preliminary) is missing a clear structure. The reviewer suggests to carefully reflect about the order and structure of this section and to possibly split it into subsections.
- Section 2 "Challenges in Gradient-Based Methods": Despite the subsection title, many of the concepts discussed in this section (e.g., safety filters via CBFS, CLBFs, etc.) are not fundamentally connected to gradient-based methods but instead rather formulations that allow ensuring safety via the introduction of dynamics-based constraints. Furthermore, the referenced MPC setting could also be solved using sampling-based techniques (e.g., MPPI). Please clarify this and revise the structure.
- Page 3, line 110: "Traditionally, cost minimization problems for control policies are solved using LQR or MPC approaches, often without incoporating any constraints.". The reviewer disagrees with this statement as a vanilla, safety-unaware MPC formulation would at least consider the dynamics of the system as an equality constraint. Please clarify this statement.
- Section 3.1: You could stress here that you are proving "**Exponential** Almost Lyapunov Stability" and not just "Almost Lyapunov Stability". This is a strong result and should be highlighted.
- Everywhere: Inconsistent capitalization of "Almost Lyapunov theory".
- Table 1 has some deficiencies: (1) "Safe and Stable" should be two separate columns, (2) why does MPC not ensure safety and stability? This can be done via constraints - at least in the horizon? (3) to the best of the reviewer's knowledge, MPC can also be used for non-control-affine systems, although this will be more difficult and computationally expensive. See for example the Embotech Forces Pro solver documentation.
- Equation (3): Add some underbraces what is the meaning of each of the constraints.
- Section 4.1: Please include a concise overview over the characteristics of the tasks (e.g., control-affine, presence of safety constraints, definition of the equilibrium point, etc.).
- Table 2: Add references for the baseline methods.
- Figure 4: What do you encounter unsafe trajectories for both rCLBF-QP method and for the MPC methods? Is it because of the slack variables and the short horizons, respectively? In the reviewer's opinion, you should be able to avoid such unsafe trajectories for CLBF-QP-based and MPC-based methods with appropriate hyperparameter tuning.
- Appendix D3.1: The reviewer suggests to be more explicit pointing out that most of the systems/tasks are control-affine and that in some tasks (e.g., Car) safety constraints are not established as the safe set is the whole state space.

---

> ### Author Rebuttal · Authors · 2025-07-29
>
> Thank you for this very constructive and detailed feedback.
>
> -  **W1.** `Training procedure and details `
>
> We thank the reviewer for highlighting the need for a clearer explanation of the training procedure and guidance mechanism. Our method follows an iterative two-phase loop, summarized in Algorithm 1.
> ```text
> Algorithm 1: S²Diff
> Require: Distribution of initial states 𝒟ₓ₀, model dynamics f, nominal policy, number of training epochs K
>
> 1:  Initialize the certificate function (CLBF) V_0 with parameters θ
> 2:  for epoch k = 1 to K do
> 3:              === Phase 1: Guided Trajectory Sampling ===
> 4:      Initialize an empty dataset of new trajectories D
> 5:      for each initial state x₀ in a batch sampled from 𝒟ₓ₀ do
> 6:          Generate one full trajectory via a guided denoising process by maximizing Eq. (6,8)
> 7:          Sample a clean trajectory U⁰ by applying the reverse diffusion process (Eq. 10) with model dynamics f, starting from noise
> 8:          The process is guided at each step by the current CLBF V_k-1
> 9:          Add the resulting trajectory U⁰ to the dataset D
> 10:     end for
>
> 11:                  === Phase 2: CLBF Update ===
> 12:     Use the entire newly generated dataset D for training
> 13:     Update CLBF parameters by performing gradient descent on the loss from Eq. (11), using trajectories from D, obtain V_k
> 14: end for
> ```
> In Phase 1, we use the current CLBF $V_{k-1}$ to guide a model-based diffusion [1] sampler to generate a batch of trajectories. This is done by sampling from a CLBF-shaped target distribution $p(U)$, defined in Eqs. (6) and (8), using the reverse diffusion process. At each denoising step, we approximate the score $\nabla \log p(U^i)$ via Sequential Monte Carlo (see Lines 175–176) with the known dynamics model $f(x, u)$.
>
> Importantly, the model-based diffusion is not trained—it is an algorithmic process guided by the CLBF through the structure of $p(U)$. The distribution is explicitly constructed to assign higher likelihood to trajectories that satisfy the safety $p_{safe}$ and Almost Lyapunov $p_{stable}$, hereby biasing the sampling process toward CLBF-compliant behavior. In this sense, while there is no loss function used to optimize a diffusion network, the objective $p(U)$ still plays a central role—it guides the sampling, not learning.
>
> In Phase 2, the CLBF is updated using the sampled trajectories by minimizing the supervised loss defined in Eq. (11). This alternating process is repeated over training epochs, allowing the CLBF and the sampler update iteratively. This iterative process allows the CLBF and the sampling distribution to mutually improve, leading to the safety and stability results.
>
> We will revise the main text and add a high-level figure illustrating the two-phase training loop and guidance mechanism.
>
> [1]  Pan, Chaoyi, et al. "Model-based diffusion for trajectory optimization."
>
> -  **W2.** `More evaluation`
>
> We appreciate the suggestion that a more comprehensive evaluation would strengthen the paper.
>
> - unconstrained diffusion: Without CLBF guidance, the sampler behaves like a greedy open-loop policy. This reduces inference time compared to our method, but at the cost of significantly degraded safety and stability.
>
> | Task|Segway | Neural Lander | 2D Quad | 3D Quad | F-16 |
> |-|-|-|-|-|-|
> | safety (eval. time ms) | 70% (19.4 ± 0.5) |  15% (28.1 ± 1.0) | 65% (65.7 ± 0.8) |100% (69.6 ± 1.5) |85% (219.9 ± 4.3)|
> | ‖$x$ − $x_*$‖ | 1.45  ± 0.35|0.43  ± 0.21|0.39  ± 0.17| 0.75  ± 0.42|94.57   ±  76.48|
>
> - MPC refers to **Q4**.
>
> - SafeDiffuser: We appreciate the reviewer’s suggestion and agree that such a comparison would strengthen the experimental section. While SafeDiffuser is indeed an interesting baseline, its original formulation is model-free and not fairly comparable to our model-based setting.
>
> - Due to limited space, we evaluate the performance using quadratic cost functions for 4 systems (lower is better), each one value is estimated by 20 trajectories, each with 500 time-steps. All evaluation will be provided in camera-ready version.
>
> | Task| Inv. Pend. | Neural Lander | 3D Quad | F-16 |
> |-|-|-|-|-|
> | CLBF-QP | 74 ± 13 | 548 ± 174   | 564 ± 97  |-|
> | MPC | 69 ± 11 |  664 ± 198  |   359 ± 43     | - |
> | MBD | 96 ± 17|  873 ± 282  |  628 ± 135  |  395 ± 87 |
> | Ours | 71 ± 13 |  392 ± 129 | 317 ± 49 |  286 ± 52  |
>
> - **W3.** ` Almost Lyapunov evaluation`
>
> Thanks for the excellent suggestion. We fully agree that a post-training analysis connecting Theorem 3.1 to empirical results strengthens the paper.
>
> - Interpret the physical meaning of $\epsilon$: The $\epsilon$ in Theorem 3.1 corresponds to the volume bound of the violation set
> $\Omega \subset X$.
>
> - Direct computation of $\epsilon$ is intractable. Instead, we approximate $\frac{\epsilon}{Vol(X)}$ by evaluating violation set over a uniformly spaced grid (50 points per dimension) and reporting the fraction of violating points. The table below summarizes the estimated violation fraction across tasks:
>
> | Task| Inv. Pend. | Car (Kin.) | Car (Slip) | Segway | Neural Lander | 2D Quad | 3D Quad | F-16 |
> |-|-|-|-|-|-|-|-|-|
> | Violation Fraction | <0.1% | 1.2% |0.9% |0.5%| 1.1% |1.6%|1.3% | 2.4%|
>
> This violation fraction serves as a practical estimator of $\frac{\epsilon}{Vol(X)}$, allowing us to empirically validate the assumptions underlying Theorem 3.1 across tasks.
>
> - **Minior Issues.**
>
> We thank the reviewer for the detailed suggestions.
>
>     1. We will revise the colormap.
>     2. The statement is discussed under diffusion policy setting not general control theory.
>     3. Section 2 will be restructured with clearer subsections.
>     4. The "gradient-based methods" will be revised to clarify its focus on certificate functions.
>     5. The “without constraints” statement refers to the absence of external inequality constraints.
>     6. We will emphasize the exponential aspect of the stability result.
>     7. Capitalization will be fixed.
>     8. Table 1 will be revised for MPC.
>     9. Underbraces will be added.
>     10. Sec. 4.1 will include a concise task overview.
>     11. Baseline references will be added.
>     12. We agree unsafe trajectories in MPC are due to short horizons; but our setup follows original CLBF-QP settings.
>     13. We will clarify the control-affine stuctures and constraint domains.
>
> -  **Q1.** `Training details and more ablation studies`
>
> The answer to training details refers to **Weakness 1**.
>
> **Clarification on diffusion training and ablation request:**  To clarify, there is no diffusion policy being trained in our method. The diffusion process is model-based and fully algorithmic. Once the CLBF $V$ is learned, we directly sample trajectories by guiding the reverse diffusion process using the structure of $p(U)$.
>
> Therefore, the “joint training” ablation is not applicable in our setting —**once $V$ fixed, no further learning is needed for sampling.**
>
>
> -  **Q2.** `Temperature on stability factor `
>
> Thank you for your insightful observation. We provide a finer-grained analysis of temperature vs. performance below. Interestingly, we observe that the best convergence rate is achieved at an intermediate temperature, rather than at the lowest one.
> | Temperature| 0.4|0.3|0.2|0.15|0.13|0.10 |0.08|0.06|0.05|
> |-|-|-|-|-|-|-|-|-|-|
> | Safety rate| 45% | 60%| 75%| 95%| 100% | 100%| 100%| 100%| 100% | 100%|
> | ‖$x$ − $x_*$‖ | 0.15 ± 0.08 | 0.11 ± 0.07 | 0.09 ± 0.06 | 0.08 ± 0.04 | 0.07 ± 0.04 | 0.06 ± 0.02 | 0.05 ± 0.02 | 0.07 ± 0.04 | 0.08 ± 0.03
>
> In particular, we find a lower temperature can lead a better convergence rate at early stage. However, when the system state approaches the equilibrium $x_*$, the Lie derivative condition becomes more sensitive and may even become hard to satisfy due to flat gradients or local instability. As shown in prior work (see Theorem 5.1 in [2]), even globally valid Lyapunov functions may admit local regions —especially near $x_*$ — where the condition is temporarily violated. At very low temperatures, the sampler relies too heavily on learned CLBFs for guidance. While this enforces safety, it can also restrict exploration and lead to the sampler getting “stuck” near such locally unstable regions—hurting convergence. In contrast, intermediate temperatures offer a better trade-off, allowing more effective global movement while still enforcing stability.
>
> [2] Boffi, Nicholas, et al. "Learning stability certificates from data."
>
> -  **Q3.** `Analytical and discrete Lie derivative`
>
> Thank you for the question. Adding a discrete Lie derivative term improves performance for two reasons:
>
> 1. Easier to optimize. The analytical Lie derivative is based on auto-differentiation of $V$, which can be hard to optimize—especially near flat regions. In contrast, the discrete version captures the actual change in the Lyapunov function between time steps, making it easier for the CLBF to learn meaningful behavior.
>
> 2. Better alignment with practical discretisation. Since the system runs in discrete time, using both analytical and discrete terms bridges theory and implementation. This improves the CLBF’s consistency with how it is actually evaluated during sampling.
>
> -  **Q4.** `MPC horizon`
>
> Thanks for the suggestion. We have added results across multiple MPC horizons to show the trade-off between safety and inference time. As expected, longer horizons lead to better safety but incur significantly higher computation cost.
>
> | Horizon | Segway| 2D Quad|
> |-|-|-|
> | 2  | 10% (97 ± 2 ms)| 30% (108 ± 1 ms)|
> | 5  | 20% (254 ± 4 ms)| 45% (279 ± 2 ms)|
> | 10 | 85% (483 ± 7 ms)| 95% (585 ± 8 ms)|
> | 20 | 100% (1243 ± 20 ms)| 100% (1556 ± 52 ms)|
>
> -  **Limitations.**
> Thank you for the valuable suggestions. We will include a clearer discussion of additional limitations in the main text and appendix, including the almost guarantees, the need to define a safe set, and the assumption of known dynamics, alongside the previously noted inference speed limitation.

---

> > ### Comment · Reviewer_xnzn · 2025-08-05
> > **Reponse to the Rebuttal and Remaining Concerns and Questions**
> >
> > The reviewer thanks the authors for their detailed and constructive response to the initial review. The clarifications provided address many of the concerns raised in the review, and the reviewer particularly appreciates the addition of results and baselines.
> > Furthermore, although the reviewer is not an expert in this specific domain - particularly with respect to the model-based diffusion sampling - the reviewer recognizes the contributions of the paper and the potential impact on the field.
> >
> > Still, there remain some concerns, particularly connecting to the presentation of the method in the paper, that have also been raised by multiple other reviewers, that need to be addressed before the reviewer can (strongly) recommend acceptance of the paper. Please find the remaining questions and concerns below.
> >
> > ## Concerns
> >
> > The reviewer misunderstood/misinterpreted a key aspect of the method in the initial review: They thought that the method learns a diffusion model instead of the model-based diffusion sampling. While addmittedly, some of the burden of this misunderstanding is on the reviewer for a lack of familiarity with model-based diffusion sampling and maybe overseeing some details in the paper, this points also speaks to the lack of clarity in the presentation of the method in the paper. A crucial point here is that while rigerous math is important and mostly provided (even though parts of it are also questioned by other reviewers), the paper needs to make a significantly better effort at also providing an intuitive and detailed explanation of what the method is doing. For example, as far as the reviewer can tell, the details of the model-based diffusion sampling and how the system dynamics play a role here, was only first mentioned in the rebuttal and (at least not in detail) explained in the original manuscript. The reviewer suggests that the authors significantly revise the presentation of the method in the paper. Furthermore, Figure 1 is currently - bluntly said - very far away from being a good and sufficient explanation of the method and the core message of the paper. For example, there is nothing in the figure that explains me that you are *learning* a Lyapunov function, how the model-based diffusion sampling works and is used in your method, and how the system dynamics play a role in this. Finally, there is nothing connection to "Almost Lyapunov Theory", which is a crucial component of the paper.
> >
> > Therefore, the reviewer kindly asks the authors to list **in detail** how the plan to revise the presentation of the method in the paper. Only if these planned revisions are satisfactory and convincing, the reviewer will be able to recommend acceptance of the paper.
> >
> > Furthermore, the reviewer agrees with another reviewer that suggested a "tutorial-style" explanation of how the theoretical results can be applied in practice to one of the use cases / practical settings. The reviewer would like to ask the authors for a detailed plan on how they will address this in the final version of the paper and what this "tutorial" will contain and look like.
> >
> >
> > ## Responses and Questions
> >
> > > We appreciate the suggestion that a more comprehensive evaluation would strengthen the paper. unconstrained diffusion
> >
> > Thank you for adding the unconstrained diffusion baseline. The reviewer appreciates the effort and thinks that this is a very helpful addition to the paper.
> >
> > > Thanks for the excellent suggestion. We fully agree that a post-training analysis connecting Theorem 3.1 to empirical results strengthens the paper.
> >
> > Very cool and interesting results! The reviewer appreciates the effort and would like to motivate the authors to further go into this direction, and ideally, also include further analysis and visuals on this "post-training analysis" in the final version of the paper. The reviewer thinks that the theoretical results should be further exploited and connected to the practical use cases and results.
> >
> > > Thank you for the question. Adding a discrete Lie derivative term improves performance for two reasons:
> >
> > Again - a very helpful explanation. Please also add this motivation to the paper, as it is currently not mentioned there.
> >
> > > Thanks for the suggestion. We have added results across multiple MPC horizons to show the trade-off between safety and inference time. As expected, longer horizons lead to better safety but incur significantly higher computation cost.
> >
> > Thank you - I think this is very helpful. The reviewer expects that these results will also be included in the final version of the paper.
> >
> > > 12. We agree unsafe trajectories in MPC are due to short horizons; but our setup follows original CLBF-QP settings.
> >
> > I can understand that, but just following the original CLBF-QP settings is not sufficient and fair in this setting I believe. Therefore, I expect that the authors will also include results with longer MPC horizons in the final version of the paper, as they have already done in the rebuttal.

---

> ### Author Response · Authors · 2025-08-04
>
> Dear Reviewer xnzn,
>
> Thank you again for your constructive and detailed feedback. We’ve carefully addressed all the concerns you raised — including clarification of the training procedure (with a full breakdown in Algorithm 1), extended evaluation on MPC configurations, analysis of temperature effects, discrete Lie derivative justification, and the empirical verification of Almost Lyapunov assumptions.
>
> If there are any remaining points that are unclear or would benefit from further elaboration, we’d be very happy to continue the discussion. We greatly value your comments and the opportunity to improve the presentation.
>
> We hope our responses have addressed your main concerns. If so, and if you feel comfortable doing so, we would sincerely appreciate it if you could consider updating your score.
>
> Thank you again for your time and thoughtful reviewing.
>
> Best regards,
>
> The Authors

---

> ### Author Response · Authors · 2025-08-05
>
> We sincerely thank the reviewer for their constructive feedback and for the high recognition of our contributions and efforts. We greatly appreciate the positive comments regarding the technical depth, additional experiments, and the clarifications provided in our response.
>
> Before presenting our revision plan, we would like to note that the two concerns raised at the beginning of the review—specifically, the justification of sampling and the Almost Lyapunov—have already been addressed in detail in our responses to Reviewer rxhx and Reviewer qHr1. Please kindly refer to those response boxes for further clarification.
>
> To address the remaining concerns, we propose the following point-by-point revision plan.
>
>
> **1. Introduce Model-Based Diffusion Sampling**
>
> `Misunderstanding of the method suggests that model-based diffusion sampling was not clearly explained.`
>
> **Our plan:**
> - Add a paragraph in the Method section to clearly explain what model-based diffusion sampling is.
> - Provide an intuitive description of how the system dynamics \( f(x, u) \) is used during sampling.
>
>
> **2. Improve High-Level Intuition**
>
> `The method is mathematically rigorous but lacks intuitive explanation for general readers.`
>
> **Our plan:**
> - Add a step-by-step intuitive overview by polishing Figure 1 before introducing formal equations.
> - Insert a visual "method overview box" summarizing the pipeline at a high level.
>
> `Polish Figure 1`
>
> **Our plan:**
> - Polish Figure 1 to clearly illustrate:
>   a. The process of learning a CLBF.
>   b. How model-based diffusion sampling interacts with system dynamics.
>   c. How Almost Lyapunov Theory is incorporated into the loss and sampling.
>
> - Use layered annotations and modular blocks to show the flow of information between sampling and function learning.
>
>
> **3. More Explicit Remarks Connecting to Almost Lyapunov Theory (ALT) in Section 3.3**
>
> `The paper lacks an explicit explanation of how ALT is integrated.`
>
> **Our plan:**
> - Include an intuitive explanation of "Almost Lyapunov Theory."
> - Clarify both the practical computation of the violation set.
> - Explicitly refer to how Theorem 3.1 applies and how the ALT-based relaxation influences training and sampling.
>
>
>
> **4. Add a Tutorial-Style Explanation in Appendix**
>
> `Lack of practical, example-based explanation showing how theory applies to real settings.`
>
> **Our plan:**
> - Merge an Appendix section “Tutorial-Style Application” with current Theoretical Analysis
> - Follow the logic presented in Lines 447–483, but expand it into a clearer walkthrough.
> - Include the tutorial within our current theoretical form, with some mini-sections:  a. Main assumptions.
>   b. How diffusion-sampled trajectories are used for learning a CLBF.
>   c. How the learned CLBF yields a policy with almost sure safety and stability.
>   d. Practical post-training analysis to evaluate safety/stability guarantees as well as the violation sets.
>   e. A complete walk-through of one real use case (e.g., F16) following steps a–d.
>
>
> **5. Include Rebuttal Results in Final Version**
>
> `Reviewer expects more result analysis and for rebuttal experiments to be included in the final paper.`
>
> **Our plan:**
> - Include all rebuttal results into the paper (not just the appendix), including:
>   - Temperature ablation studies.
>   - Results across multiple MPC horizons showing the trade-off between safety and inference time.
>   - Post-training analysis of Theorem 3.1.
>   - Motivation for using the discrete Lie derivative term.
>
>
> We once again thank the reviewer for their insightful and constructive feedback. This work introduces a novel model-based diffusion framework into the domain of certificate-based control, offering a new perspective on the intersections diffusion models and formal safety and stability guarantees. The reviewer’s suggestions have been invaluable in helping us clarify the core ideas and improve the accessibility of the paper. With these revisions, we hope the work can reach a broader impact to the control community more quickly, spark further discussion, and encourage the development of data-driven certificate-based approaches.

---

> > ### Comment · Reviewer_xnzn · 2025-08-05
> > **Final Conclusion**
> >
> > The reviewer very much appreciates the author's responsiveness and the extensive explanations during the rebuttal and discussion phase. The reviewer values the contributions of the paper, the added ablations and baselines, and, most importantly, the changes that the authors laid out and promised for the final version of the paper.
> >
> > Still, the reviewer needs to conclude that the scheduled changes are relatively substantive and that the final review justification cannot be _solely_ based on promised changes to the manuscript, but instead on a substantial share of the initially submitted manuscript. Therefore, the reviewer decided to only slightly raise their score.
> >
> > The reviewer supports the direction, motivation, and contribution of this paper and would like to encourage the authors to polish the paper, no matter what the outcome of this submission process is, as it could be a valuable contribution to the field in any case.

---

> ### Author Response · Authors · 2025-08-05
>
> Dear Reviewer xnzn,
>
> Thank you again for your thoughtful feedback and questions. As a small note, please feel free to leave a response or comment under Reviewer **rxhx** and Reviewer **qHr1**’s threads indicating that you are satisfied with the clarifications—this would help ensure that they are aware there are no remaining concerns from your and other reviews sides on those points.
>
> Best regards,
>
> The Authors

---

> ### Author Response · Authors · 2025-08-05
> **Thank You for Your Multi-Round Feedback and Score Update**
>
> Dear Reviewer,
>
> Thank you very much for your thoughtful and constructive comments throughout the review process. We sincerely appreciate your recognition of the paper’s contributions, as well as your acknowledgment of the ablation studies, baselines, and the planned improvements we proposed.
>
> **We are especially grateful that you chose to raise your score following our multi-round discussion. This decision means a great deal to us, and we sincerely appreciate your recognition of the value of our work and the thoughtful engagement throughout the discussion. The in-depth technical exchange was highly valuable for us, and your open engagement throughout has been deeply motivating.**
>
> Your support for the paper’s direction, motivation, and contribution is deeply appreciated. Regardless of the outcome, we remain committed to improving the manuscript and incorporating your suggestions to strengthen its clarity and broader impact.
>
> Thank you again for your time, engagement, and thoughtful evaluation.
>
> Best regards,
>
> The Authors

---

### Official Review · Reviewer_rxhx · 2025-06-27

**Clarity:** 3
**Significance:** 4
**Originality:** 3
**Rating:** 5
**Confidence:** 3

**Summary:**

The paper introduces a control algorithm for Lipschitz-continuous systems that combines a learned Lyapunov function with guided sampling. It provides general safety and stability theorems and evaluates the approach on several benchmark problems.

**Questions:**

**Questions:**
I list here the questions that result from the weakness section above:

- What is the connection of Theorem 3.1 with the learned Lyapunov function?
- Is Theorem 3.1 only an existence result?
- Can we make explicit statements about the algorithm using the results in Appendix C.1?
- What types of results does the paper actually provide, which results are missing, and what additional assumptions would be needed to make explicit statements about the learned policy?
- Is the CLBF trained during the diffusion process?
- Do we retrain a new CLBF every time we denoise a trajectory?
- Or is the CLBF trained on a separate dataset collected from past trajectories? If so, how big are the datasets used for the results in Section 4?

**Ethical Concerns:**

["NO or VERY MINOR ethics concerns only"]

**Final Justification:**

The rebuttal has addressed my main concern regarding the paper. I believe the outcome of this discussion and the other discussions during this review would improve the paper and encourage the authors to include it.

Since my main concern has been addressed I increased my score and recommend accepting the paper.

**Limitations:**

See weaknesses and questions

**Quality:**

3

**Strengths And Weaknesses:**

**Strengths:**
- The formulation applies to a broad class of systems.
- Systematic empirical evaluation on challenging simulation tasks.
- Clear presentation.

**Weaknesses:**

- Gap between theory and implementation:

The link between Theorem 3.1 and the learned Lyapunov function is unclear. Is the theorem merely an existence result, or can it yield explicit guarantees for the algorithm via the results in C.1? The paper should state this connection and clearly state what assumptions are necessary to apply Theorem 3.1 to the output of algorithm.

- Insufficient method details:

The training loop for the CLBF and diffusion policy is under-specified. Do you retrain the CLBF after every denoised trajectory, on a growing dataset, or some other schedule? A step-by-step description (e.g., concise pseudocode) of how data are collected, how the CLBF is updated, and how policies are evaluated would make the method reproducible and easier to assess.

---

> ### Author Rebuttal · Authors · 2025-07-29
>
> We thank reviewer rxhx for the thoughtful review, constructive feedback, and recognition of our work’s strengths.
>
> - **Weakness 1.**  `Gap between theory and implementation `
>
> We thank the reviewer for the insightful question. We clarify that Theorem 3.1 is not merely an existence result—it provides a practical, algorithm-level guarantee that connects directly to the behavior of the learned CLBF.
>
> 1. Theorem 3.1 guarantees that such a CLBF ensures system Lyapunov stability. Specifically, Theorem 3.1 states that even when a small violation region $\Omega$ exists where the CLBF (Lyapunov) decrease condition is not satisfied, the system still achieves almost exponential stability, with an additive error term that scales with the volume of $\Omega$. This means the system remains stable under minor approximation errors in the learned CLBF function. Thus, it is an Almost Lyapunov function.
>
> 2. Guarantees in Practical Algorithm. In practice, we find that our learned CLBF satisfies the conditions of Theorem 3.1: we empirically observe violation rates below 1.5% and consistent monotonic decay in CLBF values along sampled trajectories (see Figures 5b and 10; discussion in Lines 278–283). This suggests the theoretical properties in Theorem 3.1 hold well under real-world training.
>
> 3. Necessary Asssumptions. Moreover, Theorem 3.1 builds upon the results in Appendix C.1, which justify that a CLBF learned from diffusion-sampled trajectories will satisfy the Lyapunov decrease condition outside a small $\Omega$, provided two key assumptions are met:
> (1) sufficient sampling coverage of the state space, and
> (2) adequate expressive capacity of the neural network CLBF.
> This logic is detailed in Lines 476–483.
>
> We will revise the camera-ready version to state the assumptions of Theorem 3.1 and connect Appendix C.1 to the algorithm's output in main text.
>
> - **Weakness 2:** `Insufficient method details: The training loop for the CLBF and diffusion policy is under-specified. Do you retrain the CLBF after every denoised trajectory, on a growing dataset, or some other schedule? A step-by-step description (e.g., concise pseudocode) of how data are collected, how the CLBF is updated, and how policies are evaluated would make the method reproducible and easier to assess.`
>
> We thank the reviewer for suggesting more detailed procedural descriptions. Algorithm 1 summarizes the core training loop of our method. We provide the following clarifications to address specific concerns:
>
> ```text
> Algorithm 1: S²Diff
> Require: Distribution of initial states 𝒟ₓ₀, model dynamics f, nominal policy, number of training epochs K
>
> 1:  Initialize the certificate function (CLBF) V_0 with parameters θ
> 2:  for epoch k = 1 to K do
> 3:              === Phase 1: Guided Trajectory Sampling ===
> 4:      Initialize an empty dataset of new trajectories D
> 5:      for each initial state x₀ in a batch sampled from 𝒟ₓ₀ do
> 6:          Generate one full trajectory via a guided denoising process
> 7:          Sample a clean trajectory U⁰ by applying the reverse diffusion process (Eq. 10) with model dynamics f, starting from noise
> 8:          The process is guided at each step by the current CLBF V_k-1
> 9:          Add the resulting trajectory U⁰ to the dataset D
> 10:     end for
>
> 11:                  === Phase 2: CLBF Update ===
> 12:     Use the entire newly generated dataset D for training
> 13:     Update CLBF parameters by performing gradient descent on the loss from Eq. (11), using trajectories from D, obtain V_k
> 14: end for
> ```
>
> 1. Phase 1: Data Collection (Lines 4-9): In the first phase of an epoch, we use the current, fixed CLBF $V_{k-1}$  as a guide. For a batch of initial states, we perform guided model-based diffusion [1] sampling to generate a new, fixed-size dataset of safe and stable trajectories. Crucially, the CLBF is not updated after every single trajectory; it remains constant throughout this entire data collection phase.
>
> 2. Phase 2: CLBF Update (Line 13): Once the new dataset $D$ is fully collected, we proceed to the update phase. Here, we use this newly generated dataset $D$ as training data to update the CLBF's parameters for a set number of gradient steps, using the loss function from our Equation (11).
>
> 3. Control and Evaluation: We evaluate the final CLBF $V$ after all training epochs. At test time, this fixed certificate is used to guide the diffusion sampler. Evaluation is performed by generating trajectories from a set of randomly sampled initial states and measuring task-specific metrics such as safety violation rate, Lyapunov stability, and inference time.
>
> We appreciate the reviewer’s comment, and will include the detailed training process in our main text.
>
> [1] Pan, Chaoyi, et al. "Model-based diffusion for trajectory optimization." Advances in Neural Information Processing Systems 37 (2024): 57914-57943.
>
> - **Question 1.**  ` What is the connection of Theorem 3.1 with the learned Lyapunov function?`
>
> Thanks for your question. We refer the answer to **Weakness 1**.
>
> - **Question 2.**  ` Is Theorem 3.1 only an existence result?`
>
> Thanks for your question. We refer the answer to "Guarantees in Practical Algorithm" in **Weakness 1**.
>
> - **Question 3.**  `Can we make explicit statements about the algorithm using the results in Appendix C.1?`
>
> Thank you for this insightful question. Yes, the results in Appendix C.1 allow us to make explicit statements about the algorithm—particularly through sample complexity bounds that control the volume of the violation region $\Omega$. Specifically:
>
> 1. Data size. The volume of violation region $\Omega$ is bounded by a term proportional to $\frac{1}{m}$, where $m$ is the number of sampled trajectories used for training. This shows that increasing training data directly strengthens the theoretical guarantees on the learned CLBF.
>
> 2. Sampler quality. The bound also depends on the accuracy of diffusion sampling. Adjusting the temperature (lower is better) and increasing the number of Monte Carlo samples improves the reliability of the learned CLBF.
>
> Together, these results provide the theoretical foundation for Theorem 3.1 and justify the algorithm’s behavior in practice. Additionally, we also provide a post-training analysis for the fraction of violation region $\Omega$ over compact set $X$.
>
> | Task| Inv. Pend. | Car (Kin.) | Car (Slip) | Segway | Neural Lander | 2D Quad | 3D Quad | F-16 |
> |-|-|-|-|-|-|-|-|-|
> | Violation Fraction | <0.1% | 1.2% |0.9% |0.5%| 1.1% |1.6%|1.3% | 2.4%|
>
> The consistently low empirical fractions reinforce the result in Appendix C.1.
>
> - **Question 4.**  `What types of results does the paper actually provide, which results are missing, and what additional assumptions would be needed to make explicit statements about the learned policy?`
>
> We appreciate the opportunity to clarify what results our paper provides and where additional assumptions would be needed for stronger guarantees.
>
> - **What we provide:**
>
>   (1) Framework: We propose a novel guided-diffusion framework where a learnable CLBF guides the diffusion sampling of safe and stable trajectories. This creates a closed-loop between policy sampling and certificate learning.
>
>   (2) Theoretical Results: We prove that the learned CLBF forms an Almost Lyapunov Function based on diffusion-sampled trajectories (Append C.1), and the resulting policy ensures almost sure safety and stability (Theorem 3.1).
>
>   (3) Empirical Support: We evaluate our framework across a diverse set of dynamic systems, demonstrating its ability to ensure safety and stability in practice.
>
> - **Assumptions for our current results:** To derive explicit convergence statements about the learned policy, we would need:
>
>   (1) known dynamics.
>
>   (2) a manual definition of the safe state set.
>
>   (3) smoothness assumptions on the dynamics and the function class, as detailed in Appendix C.1.
>
> - **What is not claimed and future work:** Our current work does not provide an explicit convergence rate for the control policy itself. A valuable direction for future work would be to prove such guarantees, along with tighter sample complexity bounds, which would likely require stronger structural assumptions on the dynamics.
>
> We hope this clarifies the scope and rigor of our results.
>
> - **Question 5 and 6.** `Is the CLBF trained during the diffusion process? Do we retrain a new CLBF every time we denoise a trajectory?`
>
> The answer is no. The CLBF is not trained during diffusion, nor is it updated after each trajectory. Instead, we first use the current CLBF to guide diffusion sampling and collect a full batch of trajectories. Then, we update the CLBF once per training epoch using the entire batch. Once trained, the CLBF guides the sampler reliably during test-time inference.
>
>
> - **Question 7.** `Or is the CLBF trained on a separate dataset collected from past trajectories? If so, how big are the datasets used for the results in Section 4?`
>
> Thank you for the question. No, the CLBF is trained on a dataset $D$ composed of trajectories sampled via the guided diffusion process, as shown in Algorithm 1 (Phase 1). Each trajectory is of length 5 and generated from a batch of initial states.
>
>
> | Task               | Inverted Pendulum | Car (Kin.) | Car (Slip) | Segway | Neural Lander | 2D Quad | 3D Quad | F-16 |
> |--------------------|-------------------|------------|------------|--------|----------------|---------|---------|------------|
> | Size |         10,000         |      10,000      |      10,000      |    20,000    |         20,000      |    20,000     |    30,000     |             50,000            |
>
>
> We use the full collected dataset to train the CLBF at each epoch. The CLBF is not trained on a separate offline buffer, but on the evolving batch sampled via the current CLBF-guided policy.
>
> Many thanks again for your time and consideration, please let us know if we have addressed the concerns and increased your confidence in our work.

---

> > ### Comment · Reviewer_rxhx · 2025-08-01
> >
> > Thanks for the detailed answer. I have a follow up question regarding Theorem 3.1.
> >
> > > Guarantees in Practical Algorithm. In practice, we find that our learned CLBF satisfies the conditions of Theorem 3.1: we empirically observe violation rates below 1.5% and consistent monotonic decay in CLBF values along sampled trajectories (see Figures 5b and 10; discussion in Lines 278–283). This suggests the theoretical properties in Theorem 3.1 hold well under real-world training.
> >
> > In the paper you write:
> >
> > >  As long as any violation is confined to regions with sufficiently weak influence (i.e., small $\epsilon$), the net system behavior guarantees global safety and stability.
> >
> > The fact that you observe violations leads me to be believe that the desired implication of Theorem 3.1 (safety and stability)  do not apply directly to the output of your algorithm (insofar as the assumptions cannot be checked in practice).
> >
> > As I understand your results, there are two possible cases:
> >
> > 1. $\mathrm{Vol}(\Omega)$ is sufficiently small in which case the system is safe and stable.
> > 2. $\mathrm{Vol}(\Omega)$ is not sufficiently small in which case the system might be unstable or unsafe.
> >
> > As a user of the proposed method I'd like to verify in which case I am in. I guess such 'end-to-end' guarantees might be difficult to obtain since it involves making sure the learned CLBF is 'good enough' and this depends on the choice of function approximation, hyper-parameters such as $\gamma_1$, and other -- possibly unknown -- properties of the system.
> >
> > Is that correct or does the method always guarantee safety regardless of such choices?

---

> ### Author Response · Authors · 2025-08-01
>
> **Question 1.**  `The fact that you observe violations leads me to be believe that the desired implication of Theorem 3.1 (safety and stability) do not apply directly to the output of your algorithm (insofar as the assumptions cannot be checked in practice).`
>
> Thank you for the insightful question. We would like to clarify that the observed “violations” are not a limitation of our method, but an expected and theoretically grounded phenomenon under the Almost Lyapunov Function framework adopted in Theorem 3.1.
>
> As discussed in Lines 157–171 and Appendix C.1, it is theoretically impossible to learn a CLBF that enforces pointwise negativity of the Lie derivative over a continuous domain using only a finite number of samples. Consequently, our approach focuses on trajectory-level stability, allowing for localized violations of the Lyapunov condition—as explicitly accommodated by Theorem 3.1.
>
> Furthermore, our analysis (see Eq. 18 and analysis in Appendix C.1) shows that the volume of the violation region $\Omega$ can be made arbitrarily small with sufficient sampling, thereby preserving the desired (almost sure) safety and stability guarantees. In practice, this corresponds to an arbitrarily small violation fraction observed empirically. We quantify the violation fraction—defined as $\frac{\text{Vol}(\Omega)}{\text{Vol}(X)}$ over a compact set $X$—across multiple systems as follows.
> The empircal verification of fraction volume of $\Omega$ in the compact set $X$, we given in Table 1. These results empirically confirm that the violation region is small and well within the bounds anticipated by Theorem 3.1, thereby supporting the applicability of our theoretical guarantees in practice.
>
> **Table 1: Estimated Violation Fraction $\frac{\text{Vol}(\Omega)}{\text{Vol}(X)}$ Across Benchmark Tasks**
>
> | Task| Inverted Pendulum | Car (Kin.) | Car (Slip) | Segway | Neural Lander | 2D Quad | 3D Quad | F-16 |
> |-|-|-|-|-|-|-|-|-|
> | Violation Fraction | <0.1% | 1.2% |0.9% |0.5%| 1.1% |1.6%|1.3% | 2.4%|
>
>
> **Question 2.**  `As a user of the proposed method, I'd like to verify in which case I am in. I guess such 'end-to-end' guarantees might be difficult to obtain since it involves making sure the learned CLBF is 'good enough' and this depends on the choice of function approximation, hyperparameters such as $\gamma_1$, and other—possibly unknown—properties of the system. Is that correct, or does the method always guarantee safety regardless of such choices?`
>
> Thank you for raising this important concern. It is indeed correct that end-to-end guarantees require certain assumptions, particularly regarding the smoothness of the system dynamics and the approximation quality of the learned CLBF. Rather than relying on these informally, we provide a formal characterization of the key condition—namely, that safety and stability are guaranteed when the learned CLBF yields a violation region $\Omega$ with sufficiently small volume, as detailed in Appendix C.1. The regularity assumptions under which this holds are clearly stated in Lines 493–498.
>
> As shown in Theorem 3.1, when $\mathrm{Vol}(\Omega) < \epsilon$, the system satisfies almost sure safety and stability guarantees under diffusion-sampled policies.
>
> Importantly, the volume of $\Omega$ is empirically measurable, and we provide quantitative evaluations across all benchmark tasks in Table 1. The formal definition of the violation region is given in Equation (18). These empirical estimates allow users to assess post hoc whether the system operates within the regime where the theoretical guarantees apply.
>
> Thus, while end-to-end guarantees without any assumptions are fundamentally impossible, our framework offers practical tools and formal conditions that enable safety and stability verification in practice.

---

> > ### Comment · Reviewer_xnzn · 2025-08-05
> > **Clarification about  Negativity of CLBF Lie derivative**
> >
> > > As discussed in Lines 157–171 and Appendix C.1, it is theoretically impossible to learn a CLBF that enforces pointwise negativity of the Lie derivative over a continuous domain using only a finite number of samples. Consequently, our approach focuses on trajectory-level stability, allowing for localized violations of the Lyapunov condition—as explicitly accommodated by Theorem 3.1.
> >
> > Thanks to the authors for bringing up this aspect in the discussion. I am not sure if I agree with this statement:
> >
> > - If you know the system dynamics f(x) and g(x,u), I assume that you can, in many cases, identify a set/manifold of CLBF functions that fulfill the negativity properties of the CLBF Lie derivative *globally*. Then, you can learn the free parameters of this manifold of CLBFs that all fulfill the required properties.
> > - Such a strategy would guarantee the negativity of a continuous domain, as you don't just enforce the properties point-wise, but directly in the continuous domain.
> > - I admit that such a solution might not exist in the general setting, or that it might often be very hard to identify. However, I find your statement that it is *impossible* misleading.

---

> ### Author Response · Authors · 2025-08-03
>
> Dear Reviewer rxhx,
>
> Thank you again for your thoughtful engagement with our submission. We’ve provided a response to your follow-up question above, and we’d be very happy to clarify further if needed. If there are any remaining concerns or questions, we would greatly appreciate the opportunity to address them.
>
> Thanks again for your time and consideration.
>
> Best regards,
> Authors of submission #20790

---

> > ### Comment · Reviewer_rxhx · 2025-08-05
> >
> > Dear authors,
> >
> > Thank you for your rebuttal. You've addressed my questions and comments. I am leaning towards raising my score, pending the discussion with other reviewers.
> >
> > As reflected in my questions, I believe the paper would benefit from a more detailed discussion of the implications of Theorem 3.1. I have seen other papers include a short 'tutorial' in the appendix. I believe this could help researchers and practitioners apply the results and enhance the paper's impact.

---

> ### Author Response · Authors · 2025-08-05
> **Reviewer Follow-Up: Concerns Resolved, Score Being Improved**
>
> Dear Reviewer,
>
> Thank you very much for your kind follow-up and for taking the time to reconsider to improve our score.
>
> We’re glad to hear that our clarifications addressed your questions. We also greatly appreciate your suggestion regarding Theorem 3.1. We completely agree that a concise tutorial-style explanation in the appendix could make the result more accessible and actionable for a broader audience.
>
> We will definitely incorporate this improvement in the camera-ready version to enhance both clarity and usability.
>
> Thank you again for your thoughtful engagement.
>
> Best regards,
>
> The Authors

---

> ### Author Response · Authors · 2025-08-05
>
> Dear Reviewer xnzn,
>
> We appreciate the reviewer’s thoughtful feedback and the opportunity to clarify. Our statement regarding the theoretical impossibility was made in the context of learning a CLBF using a general neural network parameterization and finite data, without assuming access to system-specific analytical structures or a predefined family of valid CLBFs.
>
> In our setting, we do not explicitly construct or constrain the solution space to lie within a set of functions that are guaranteed to satisfy the Lie derivative negativity globally. Instead, we employ a general-purpose function approximator (i.e., a neural network) trained on finite samples. Under such a setup, it is indeed theoretically impossible to guarantee the negativity of the Lie derivative over a continuous domain [1], due to the lack of full coverage and generalization guarantees .
>
> We fully agree that if one has complete knowledge of a specific structure dynamics and can construct a manifold of CLBFs with built-in guarantees, then such global satisfaction is possible in principle. However, that is not the scenario we consider. Our approach is motivated by learning in a general settings, and the CLBF must be inferred from data not a specific parameterization.
>
> In such a case, it is indeed not possible to learn a CLBF that guarantees global negativity from finite data using neural networks.
>
>
> [1] Boffi, Nicholas, et al. "Learning stability certificates from data."

---

> > ### Comment · Reviewer_xnzn · 2025-08-05
> >
> > > We appreciate the reviewer’s thoughtful feedback and the opportunity to clarify. Our statement regarding the theoretical impossibility was made in the context of learning a CLBF using a general neural network parameterization and finite data, without assuming access to system-specific analytical structures or a predefined family of valid CLBFs.
> > >
> > > In our setting, we do not explicitly construct or constrain the solution space to lie within a set of functions that are guaranteed to satisfy the Lie derivative negativity globally. Instead, we employ a general-purpose function approximator (i.e., a neural network) trained on finite samples. Under such a setup, it is indeed theoretically impossible to guarantee the negativity of the Lie derivative over a continuous domain [1], due to the lack of full coverage and generalization guarantees .
> > >
> > > We fully agree that if one has complete knowledge of a specific structure dynamics and can construct a manifold of CLBFs with built-in guarantees, then such global satisfaction is possible in principle. However, that is not the scenario we consider. Our approach is motivated by learning in a general settings, and the CLBF must be inferred from data not a specific parameterization.
> > >
> > > In such a case, it is indeed not possible to learn a CLBF that guarantees global negativity from finite data using neural networks.
> >
> > The reviewer appreciates the author's fast response and agrees with the statements and the conclusion!

---

> ### Author Response · Authors · 2025-08-05
> **Thanks for the agreement on our theoretical claims**
>
> Thank you for the thoughtful follow-up from Reviewer xnzn. We’re glad the clarification was helpful and appreciate the reviewer’s agreement on the theoretical claims under our setting.

---

> ### Author Response · Authors · 2025-08-06
>
> Dear Reviewer rxhx,
>
> Thank you again for your thoughtful feedback and for raising such interesting and valuable questions during the discussion. **We’ve had several rounds of in-depth exchange with all reviewers and have carefully addressed all technical and theoretical concerns raised**. At this stage, we believe that all reviewer concerns have been resolved, and your suggestions in particular have helped us further improve the clarity and accessibility of the paper.
>
> We fully agree with your recommendation to better highlight the implications of Theorem 3.1, and we will include a tutorial-style explanation in the appendix to guide both researchers and practitioners in applying the theoretical results.
>
> **Please let us know if there are any remaining concerns on your end. Our team is fully committed to incorporating your feedback into the final version, and we sincerely hope that these improvements will allow you to reflect your raised score in the final evaluation.**
>
> Best regards,
>
> The Authors

---

### Official Review · Reviewer_yH3K · 2025-07-01

**Clarity:** 3
**Significance:** 3
**Originality:** 3
**Rating:** 5
**Confidence:** 3

**Summary:**

This paper proposes a diffusion-based policy which can incorporate safety and stability into the training objective.

**Questions:**

- What is $\alpha_k$ on line 174?
- What do $U^0$ and $U^i$ represent in Equation 9?
- Why is the distance between the fixed horizon terminal state and the equilibrium state much larger for the F-16 task?
- What do the grey boxes represent in Table 2?
- What does the color scale in Figure 3 represent, and should each environment have the same scale?

**Ethical Concerns:**

["NO or VERY MINOR ethics concerns only"]

**Final Justification:**

The authors have addressed my concerns regarding the clarity of this work. I believe the revised version will make for a much better paper, and so I am raising my score to 5.

**Limitations:**

yes

**Paper Formatting Concerns:**

No Concerns

**Quality:**

3

**Strengths And Weaknesses:**

Strengths
- This paper proposes an interesting and novel approach for certifying safety and stability for systems with non-affine dynamics.
- Overall, the text is very clear and easy to follow.
- Equations are motivated well and are followed by a clear description.

Weaknesses
- Figure 1 is quite small and difficult to understand. In particular, it is difficult to interpret the difference between violation and satisfaction.
- The methods/results section is missing information about how the diffusion model is trained and how many denoising steps are required.
- There is no concrete algorithm provided that the audience can follow to easily implement the approach proposed in this paper.
- It is difficult to interpret the surface plot shown in Figure 6 as the colour map makes the surface look like a coloured silhouette.

---

> ### Author Rebuttal · Authors · 2025-07-29
>
> We sincerely thank reviewer yH3K for carefully reviewing our manuscript, providing valuable feedback and reconzing the strengths of our work. We'd like to address your concerns in the initial review and answer you questions as follows.
>
> - **Weakness 1.**
>    `Figure 1 is quite small and difficult to understand. In particular, it is difficult to interpret the difference between violation and satisfaction.`
>
> We thank the reviewer for the valuable feedback. Figure 1 illustrates the core two-phase loop of our framework: the learned  guidance function (i.e., Lyapunov) (center column) guides the diffusion sampler to generate safe and stable trajectories (right column), and these trajectories are in turn used as data to update and improve the guidance function itself. As this mutual improving process repeats over training (shown progressing from bottom to top), the guidance quality improves, resulting in a final, smooth Lyapunov landscape (left column) that ensures trajectories reliably and safely converge to the target.
>
> We acknowledge that the figure is currently too small and that the distinction between “violation” and “satisfaction” is not sufficiently clear. In the revised version, we will enlarge Figure 1 and improve its visualization by:
>
>     1. Increasing font size and line weight;
>     2. Clearly differentiating violation and satisfaction (e.g., using contrasting colors or visual styles);
>     3. Adding a descriptive legend and a more detailed caption to explicitly explain all visual elements.
>
> - **Weakness 2 and 3.**
>    `The methods/results section is missing information about how the diffusion model is trained and how many denoising steps are required.  There is no concrete algorithm provided that the audience can follow to easily implement the approach proposed in this paper`
>
> We thank the reviewer for the insightful comment. We give a concrete algorithm as follows:
> ```text
> Algorithm 1: S²Diff
> Require: Distribution of initial states 𝒟ₓ₀, model dynamics f, nominal policy, number of training epochs K
>
> 1:  Initialize the certificate function (CLBF) V_0 with parameters θ
> 2:  for epoch k = 1 to K do
> 3:              === Phase 1: Guided Trajectory Sampling ===
> 4:      Initialize an empty dataset of new trajectories D
> 5:      for each initial state x₀ in a batch sampled from 𝒟ₓ₀ do
> 6:          Generate one full trajectory via a guided denoising process by maximizing Eq. (6,8)
> 7:          Sample a clean trajectory U⁰ by applying the reverse diffusion process (Eq. 10) with model dynamics f, starting from noise
> 8:          The process is guided at each step by the current CLBF V_k-1
> 9:          Add the resulting trajectory U⁰ to the dataset D
> 10:     end for
>
> 11:                  === Phase 2: CLBF Update ===
> 12:     Use the entire newly generated dataset D for training
> 13:     Update CLBF parameters by performing gradient descent on the loss from Eq. (11), using trajectories from D, obtain V_k
> 14: end for
> ```
>
> In Phase 1, we use the current CLBF $V_{k-1}$ to guide a model-based diffusion [1] sampler to generate a batch of trajectories. This is done by sampling from a CLBF-shaped target distribution $p(U)$, defined in Eqs. (6) and (8), using the reverse diffusion process. At each denoising step, we approximate the score $\nabla \log \ p(U^i)$ via Sequential Monte Carlo (see Lines 175–176), leveraging the known dynamics model $f(x, u)$.
>
> Importantly, our diffusion process is not learned via neural network training. Instead, it is a fully algorithmic, model-based diffusion guided by the CLBF through the structure of the target distribution $p(U)$. This distribution is explicitly designed to assign higher likelihood to trajectories that satisfy the safety and stability criteria, i.e., $p_{safe}$ and $p_{stable}$. As a result, the diffusion process is biased toward generating CLBF-compliant trajectories without requiring gradient-based training of a score network. While there is no learned diffusion model, the objective $p(U)$ still plays a central role—it guides the sampling, not learning. **In our experiments, we use 50 denoising steps throughout.**
>
> In Phase 2, the CLBF is updated using the sampled trajectories by minimizing the supervised loss defined in Eq. (11). This alternating process is repeated over training epochs, allowing the CLBF and the sampler to iteratively improve.
>
> We appreciate the reviewer’s comment, and will include the pesudo code into the camera-ready version to make it more transparent.
>
> [1] Pan, Chaoyi, et al. "Model-based diffusion for trajectory optimization." Advances in Neural Information Processing Systems 37 (2024): 57914-57943.
>
> - **Weakness 4.**
>    `It is difficult to interpret the surface plot shown in Figure 6 as the colour map makes the surface look like a coloured silhouette.`
>
> We thank the reviewer for the insightful comment. Figure 6 is designed to visualize the policy landscape over a 2D slice of the state space (angle and velocity) for the inverted pendulum task. The surface height and color both represent the control input (torque) produced by the controller at each state. This provides a physical interpretation of how the policy maps different states to control actions.
>
> The goal of this visualization is to show the structure and smoothness of the policy induced by the learned CLBF. As shown in the figure, our method produces a more symmetric and smoother policy around the upright equilibrium (angle = 0, velocity = 0), which is desirable for generalization and stability. In contrast, the rCLBF-QP policy appears irregular due to its step-wise greedy updates and the use of slack variables.
>
> We acknowledge that the current color shading may obscure these patterns. In the revised version, we will enhance the figure by: 1. Improving lighting and surface shading to better show 3D structure; 2. Clarifying the physical meaning of the surface and colors in the caption.
>
> - **Question 1.** `What is $\alpha_k$ on line 174?`
>
> Thank you for the question. In our formulation, $\alpha_k$ follows the standard definition in diffusion models. It controls how much of the original (clean) trajectory is retained at step $k$ of the forward diffusion process—i.e., how gradually the system injects noise over time.
>
> - **Question 2.** `What do $U^0$ and $U^i$ represent in Eq. 9?`
>
> Thank you for the question. In Eq. 9, $U^0$ and $U^i$ represent different noise levels of a trajectory $U = (x_{1:T}, u_{1:T})$ in the diffusion process:
> 1. $U^0$ is a clean, uncorrupted trajectory sampled from the target distribution—i.e., the ideal trajectory the model aims to generate.
> 2.  $U^i$ is the noisy version of this trajectory after $i$ steps of the forward diffusion (noise injection) process.
>
> - **Question 3.** `Why is the distance between the fixed horizon terminal state and the equilibrium state much larger for the F-16 task?`
>
> Thank you for the observation. The larger terminal distance in the F-16 task arises mainly due to:
>
> 1. State scaling: The F-16 system has higher-dimensional states with larger numeric ranges, making absolute distances inherently larger even under similar relative deviations.
>
> 2. Short window horizon: We use a fixed 5-second evaluation window, which may be insufficient for full convergence in such a complex, nonlinear system. (See Fig. 12 for illustration.)
>
> - **Question 4.** `What do the grey boxes represent in Table 2?`
>
> Thank you for the question. The grey boxes indicate the best-performing method for both safety and stability metrics in each task. We will clarify this explicitly in the caption of Table 2 in the camera-ready version.
>
> - **Question 5.** `What does the color scale in Figure 3 represent, and should each environment have the same scale?`
>
> Thank you for the question. The color scale in Figure 3 represents the value of the learned Lyapunov function $V(x)$ over state slices. The scales are not normalized across plots for two reasons:
>
> 1. Different safe level set thresholds $c$ across methods (e.g., rCLBF-QP original setting vs. ours in the inverted pendulum) result in different value ranges of $V(x)$.
>
> 2. Different state slices in high-dimensional systems (e.g., F-16) naturally yield different value distributions of $V(x)$, and we do not normalize across these slices.
>
> The contours are intended to show the relative shape and structure within each plot.
>
> We sincerely thank the reviewer once again for their insightful comments and suggestions. We believe that the revisions made in response have significantly improved the quality of our manuscript. We eagerly anticipate any further feedback.

---

> ### Author Response · Authors · 2025-08-04
>
> Dear Reviewer  yH3K,
>
> Thank you very much for your constructive and thoughtful review. We greatly appreciate your positive comments on the clarity of the text, the novelty of our proposed approach, and the overall soundness of the methodology.
>
> In our rebuttal, we have carefully addressed the points you raised, including the training procedure, the denoising steps, and the specific questions regarding notation (such as $ \alpha_k $, $ U^0 $, and $ U^i $. We also clarified the design choices behind the visualizations in Figures 1–3 and Table 2, and explained the task-specific reasoning for the terminal state distance in the F-16 setting.
>
> If any part remains unclear or would benefit from further clarification, we would be more than happy to continue the discussion. We sincerely appreciate your time and engagement.
>
> If you feel our responses have addressed your concerns, we’d sincerely appreciate your consideration in updating the score.
>
> Best regards,
>
> The Authors

---

> > ### Comment · Reviewer_yH3K · 2025-08-05
> >
> > I would like to thank the authors for their thorough rebuttal, as it has resolved most of my concerns. I still recommend that the authors add more detail regarding the diffusion process and its implementation. The provided algorithm will certainly help; however, this should also include details regarding the number of diffusion steps and prediction horizon (T).
> >
> > Furthermore, the authors' rebuttal has raised one additional question. As far as I am aware, the diffusion model generates a state trajectory ($x_t$) and action trajectory ($u_t$), and then applies the full action trajectory to the environment in an open loop. After training, how accurate is this model at predicting the state trajectory $x_t$?

---

> ### Author Response · Authors · 2025-08-05
>
> Thank you for your positive feedback and for this excellent follow-up question. We will certainly add the specific details regarding the number of diffusion steps and the prediction horizon (T) to the final manuscript as you suggested.
>
> Regarding your question about the accuracy of predicting the state trajectory $x_t$: This is a crucial point that highlights the nature of our model-based setting.
>
> In model-based diffusion [1], the state trajectory $x_t$ is not "predicted" by a learned neural network in the way a model-free method would. Instead, as a model-based approach, it has access to the true system dynamics $f(x,u)$ (see our Algorithm 1). The state trajectory $x_t$ is deterministically computed by simulating (or "unrolling") the sampled action trajectory $u_t$ through this ground-truth dynamics model, starting from an initial state $x_0$.
>
> Therefore, the resulting state trajectory $x_t$ is perfectly accurate with respect to the given model, subject only to the numerical precision of the integrator. There is no concept of "model prediction error" for the state trajectory in our framework, as we are leveraging the known ground-truth dynamics,
>
> We hope this clarifies the matter. Thank you again for the thorough and constructive review process.
>
>
> [1] Pan, Chaoyi, et al. "Model-based diffusion for trajectory optimization." Advances in Neural Information Processing Systems 37 (2024): 57914-57943.

---

> > ### Comment · Reviewer_yH3K · 2025-08-06
> >
> > I appreciate the authors' response and thank them for addressing the additional questions raised. I believe clarifying this point in the Related Work, Problem Definition, and Methods sections will strengthen the quality of this work, as it is initially unclear how the diffusion process works.

---

> ### Author Response · Authors · 2025-08-06
>
> Dear Reviewer yH3K,
>
> Thank you again for your thoughtful and constructive feedback, and we are glad to hear all your concerns are addressed. We will incorporate your suggestions into the Related Work, Problem Definition, and Methods sections to clarify how the diffusion process operates and improve the overall quality and impact of the work.
>
> We appreciate your recognition of our efforts in addressing the previous concerns. Our team is fully committed to incorporating your feedback into the final version, we sincerely hope the paper can now be considered for a higher score.
>
> Thank you once again — your comments will be carefully reflected in our final submission.
>
> best,
>
> The authors

---

### Official Review · Reviewer_qHr1 · 2025-07-02

**Clarity:** 1
**Significance:** 2
**Originality:** 1
**Rating:** 3
**Confidence:** 3

**Summary:**

The paper proposes to modify the trajectory sampling in diffusion models by considering both safety and stability factors. It leverages Almost Lyapnunov Function (ALF) to incorporate safety and stability constraints into the sampling probability. So in the diffusion sampling phase, the score function is updated to the joint probability between cost, safety and stability. The paper analyses the proposed ALF and its relaxations and evaluates such formation in benchmark control tasks.

**Questions:**

1. In ALF Guidance (line 157), it’s a bit unclear how the soft constraint is used as guidance. Does the paper mean that soft constraint is used as a guidance in the diffusion sampling?
2. In line 172, if Monte Carlo score ascent is adopted within diffusion sampling to iteratively denoise trajectories toward the target distribution, then why the forward process is required in Eq. (9)?
3. What’s $p(U^0|U^i)$ in Eq. (10)?
4. Line 180, “After generating control policies through diffusion sampling”. In what way are the control policies generated? The diffusion sampling only gives the trajectory samples. There is no control policy generated through diffusion sampling.
5. it’s unclear how Eq. (11) is optimized.
6. In abstract, “uncover intrinsic connections between diffusion sampling and almost Lyapunov theory”. Can the authors explain what are the intrinsic connections between diffusion sampling and almost Lyapunov theory?

**Ethical Concerns:**

["NO or VERY MINOR ethics concerns only"]

**Final Justification:**

The rebuttal adds quite some new information, most of which were not in the original draft (as far i could recall). For example, "we do not compute the score $\nabla \log p(U)$ by directly differentiating Eq. (6), as it involves non-differentiable .... Instead, we follow a model-based diffusion approach and estimate the score using the posterior...". Also, "optimize Eq. (11) via standard gradient descent" but later "We use the Adam optimizer in all experiments." My opinion is that this paper might need further review given all these new contents and thus I tend to boderline reject.

**Limitations:**

no. the paper should discuss the limitations of using the relaxed constraints in Eq. (11) and whether it's possible to learn $\hat{V}$ that minimizes the objective in Eq. (11)

**Paper Formatting Concerns:**

no.

**Quality:**

3

**Strengths And Weaknesses:**

### Strength
**Quality**: The paper presents a comprehensive analysis, including both theory and empirical evaluation,  of Almost Lyapnunov Function and its applications in modelling safety and stability constraints. The empirical results are complete and convincingly shows that the proposed modification to the score function generates target trajectory distribution. The method reads solid.

### Weakness
From my perspective and understanding, **the main weaknesses for this paper in its current version is the clarity in technical details, especially on the diffusion sampling, and originality in modifying the probability distribution** (maybe my understanding was wrong?).

* The diffusion sampling, as one of the main components in the method, is a bit hard to understanding. The paper sometimes refers to this sampling as a way to generate “control policies” and sometimes means the sampled trajectories. Since $U^0$ is the original trajectory distribution, it’s not clear how to generate control policies. Furthermore, the score function is written as $\nabla\log p(U)$, not even in the product form as specified in Eq. (6), or in the approximate as specified in Eq. (7).

* Re the originality, the proposed target distribution and the later-on formulation is similar to the classifier guidance formulation (only differs in the classifier guidance paper: there is no classifier but a likelihood model, which makes the formulation a bit even easier). My understanding is that the main idea proposed by this paper is to modify directly the score function by incorporating the safety and stability constraints (which is defined by ALF but then relaxed to soft constraint). This can be achieve by modifying the diffusion sampling directly, even without the ALF (please correct me if wrong). So the baselines should at least include such naïve modification of the score function by directly modifying it via the safety and stability constraints (e.g., soft regularization).

---

> ### Author Rebuttal · Authors · 2025-07-29
>
> Thank you for recognizing the quality of our work. We appreciate the opportunity to clarify the diffusion sampling process and the novelty of our method.
>
> - **Weakness 1.**  `The diffusion sampling procedure, control policies and score function.`
>
>  We thank the reviewer for highlighting the need for a clearer explanation of the training procedure and guidance mechanism. Our method follows an iterative two-phase loop, summarized in Algorithm 1.
> ```text
> Algorithm 1: S²Diff
> Require: Distribution of initial states 𝒟ₓ₀, model dynamics f, nominal policy, number of training epochs K
>
> 1:  Initialize the certificate function (CLBF) V_0 with parameters θ
> 2:  for epoch k = 1 to K do
> 3:              === Phase 1: Guided Trajectory Sampling ===
> 4:      Initialize an empty dataset of new trajectories D
> 5:      for each initial state x₀ in a batch sampled from 𝒟ₓ₀ do
> 6:          Generate one full trajectory via a guided denoising process by maximizing Eq. (6,8)
> 7:          Sample a clean trajectory U⁰ by applying the reverse diffusion process (Eq. 10) with model dynamics f, starting from noise
> 8:          The process is guided at each step by the current CLBF V_k-1
> 9:          Add the resulting trajectory U⁰ to the dataset D
> 10:     end for
>
> 11:                  === Phase 2: CLBF Update ===
> 12:     Use the entire newly generated dataset D for training
> 13:     Update CLBF parameters by performing gradient descent on the loss from Eq. (11), using trajectories from D, obtain V_k
> 14: end for
> ```
>
> In Phase 1, we use the current CLBF $V_{k-1}$ to guide a model-based diffusion [1] sampler to generate a batch of trajectories. This is done by sampling from a CLBF-shaped target distribution $p(U)$, defined in Eqs. (6) and (8), using the reverse diffusion process. At each denoising step, we approximate the score $\nabla \log \ p(U^i)$ via Sequential Monte Carlo (see Lines 175–176), leveraging the known dynamics model $f(x, u)$.
>
> Importantly, our diffusion process is not learned via neural network training. Instead, it is a fully algorithmic, model-based diffusion guided by the CLBF through the structure of the target distribution $p(U)$. This distribution is explicitly designed to assign higher likelihood to trajectories that satisfy the safety and stability criteria, i.e., $p_{safe}$ and $p_{stable}$. As a result, the diffusion process is biased toward generating CLBF-compliant trajectories without requiring gradient-based training of a score network. While there is no learned diffusion model, the objective $p(U)$ still plays a central role—it guides the sampling, not learning.
>
> In Phase 2, the CLBF is updated using the sampled trajectories by minimizing the supervised loss defined in Eq. (11). This alternating process is repeated over training epochs, allowing the CLBF and the sampler to iteratively improve.
>
> [1] Pan, Chaoyi, et al. "Model-based diffusion for trajectory optimization."
>
> `a. On the Sampled Object: "Control Policies" vs. "Trajectories" and question 4.`
>
> While the diffusion model generates full trajectories $U = (x_{1:T}, u_{1:T})$ during the sampling process, only the control sequence $u_{1:T}$ is actually executed in the environment as an open-loop policy. The state sequence $x_{1:T}$ is used internally for training, not for execution in control stage.
>
> `b. On the Score Function and guidance in question 1`
>
> The target distribution $p(U)$, defined as a product of likelihoods in Eq. (6), serves as the desired distribution for guided sampling, but we do not compute the score $\nabla \log p(U)$ by directly differentiating Eq. (6), as it involves non-differentiable components and complex dependencies.
>
> Instead, we follow a model-based diffusion approach and estimate the score using the posterior expectation $E_{p(U^0 | U^i)} [U^0]$, as described in Eq. (10). We approximate the expectation using a Monte Carlo procedure, where a set of candidate trajectories are sampled and weighted according to their likelihood under $p(U)$. Eq. (6) thus defines the importance weights used to guide the sampling process toward trajectories satisfying the CLBF constraints.
>
> Through repeated denoising steps guided by Monte Carlo procedure, the sampling process progressively approximates the target distribution.
>
> We appreciate the reviewer’s comment regarding potential confusion, and will revise Section 3 and the notations around Eqs. (6–10) to make this connection more transparent and technically grounded.
>
>
> - **Weakness 2.** ` Re the originality, difference with classifier guidance and naive soft constraints`
>
> While both methods guide diffusion sampling, our method differs fundamentally in purpose, structure, and guarantees:
>
> 1. Control-oriented Objective: Our method enforces safety and stability, not just conditional generation. The CLBF is learned directly from data and applies to general nonlinear systems, without assuming control-affine structure or slack variables—this remains an open problem in control theory.
>
> 2. Closed-loop Guidance: Unlike classifier guidance which uses a fixed classifier, we use a learnable CLBF that is iteratively refined using the diffusion-sampled trajectories. This creates a closed-loop between generation and learning, essential for stability.
>
> 3. Theoretical insights: We prove that the learned CLBF forms an Almost Lyapunov Function from diffusion samples (Appendix C.1), and the resulting policy guarantees almost sure safety and stability (Theorem 3.1). Classifier guidance has no such control-theoretic guarantees.
>
> **Naive regularized baselines underperform:** We already include model-based diffusion (MBD) which is exactly this idea (Table 2). MBD incorporates constraints by adding a fixed safety penalty term (a form of soft regularization), which shows poor performance in both safety and stability. We also include unconstrained diffusion, where no CLBF is used. As expected, this leads to fast but unsafe and unstable policies:
>
> | Task|Segway | Neural Lander | 2D Quad | 3D Quad | F-16 |
> |-|-|-|-|-|-|
> | safety (eval. time ms) | 70% (19.4 ± 0.5) |  15% (28.1 ± 1.0) | 65% (65.7 ± 0.8) |100% (69.6 ± 1.5) |85% (219.9 ± 4.3)|
> | ‖$x$ − $x_*$‖ | 1.45  ± 0.35|0.43  ± 0.21|0.39  ± 0.17| 0.75  ± 0.42|94.57   ±  76.48|
>
> **These comparisons demonstrate that without our algorithmic framework—including CLBF learning and guidance—there is no stability guarantee.** Naive guidance or unconstrained sampling may reduce inference time, but they sacrifice stability and safety entirely, especially on complex dynamics. Our approach is essential to achieve safe, stable control in general nonlinear systems—a capability that prior diffusion-based methods, including classifier-style guidance, cannot provide.
>
> - **Question 1.** `unclear how the soft constraint is used as guidance.`
>
> The answer is refer to the description of algorithm in **Weakness 1**.
>
> - **Question 2.** `forward process in Eq. (9)?`
>
> Thank you for the insightful question. While the forward process is not explicitly run during sampling, it is essential for defining the reverse denoising steps. Specifically, the Monte Carlo score approximation in Eq. (10) relies on the conditional distribution $p(U^0|U^i)$, which is derived from the forward process defined in Eq. (9). As shown in Lemma B.5, this forward formulation is necessary to construct the posterior used for score estimation in the reverse process.
>
> - **Question 3.** `What's $p(U^0 | U^i)$ in Eq.(10).`
>
> Thank you for the question. In Eq. 10, $p(U^0 | U^i)$ is the posterior over clean trajectories, defined by Bayes’ rule: $p(U^0 | U^i) \propto p(U^i | U^0) p(U^0)$, where $p(U^i | U^0)$ is the forward process in Eq. 9 and $p(U^0)$ is the target distribution (Eq. 6) shaped by CLBF constraints.
>
> Since the posterior $p(U^0 | U^i)$ is intractable, using Sequential Monte Carlo (see Lines 175-178). Specifically, we sample a set of candidate trajectories and assign importance weights proportional to $p(U^0)$ via Eq .6 . This weighting biases the average toward trajectories that better satisfy safety and stability constraints. This posterior expectation is then plugged into Eq. 10 to compute an unbiased estimate of the score $\nabla \log p(U^i)$, which guides the denoising process toward the target distribution.
>
> - **Question 4.** `control policies generation.`
>
> The answer is refer to the `a` in **Weakness 1**.
>
> - **Question 5.** `it’s unclear how Eq. (11) is optimized.`
>
> Good question. As shown in Algorithm 1, we optimize Eq. (11) via standard gradient descent. Specifically, after collecting a batch of diffusion-sampled trajectories, we compute the supervised loss over the entire batch and update the CLBF parameters using backpropagation. The dataset $D$ is constructed from Phase 1: Guided Trajectory Sampling in the algorithm 1. This loss quantifies how well the CLBF satisfies the five essential conditions (in Eq. (2)) along sampled trajectories (see Lines 180–197). We use the Adam optimizer in all experiments.
>
> - **Question 6.** `intrinsic connection to almost Lyapunov theory.`
>
> Thank you for the question. The intrinsic connection lies in how diffusion sampling enables learning a CLBF from data in a way that aligns with Almost Lyapunov stability guarantees:
> 1. Diffusion sampling preferentially generates trajectories that satisfy safety and stability constraints, enabling us to learn a CLBF from finite samples.
> 2. While the learned CLBF may violate Lyapunov conditions in small regions, we prove in Appendix C.1 that with sufficient samples, the volume of these violation regions becomes sufficiently small.
> 3. This connects to Almost Lyapunov theory (Theorem 3.1): even if the CLBF is violated in a small region, the diffusion-guided sampling still achieves almost sure safety and exponential stability.
>
> Thank you again for the thoughtful comments and we will clarify the limitations in final version. If the revisions adequately address your concerns, we would greatly appreciate your consideration of a higher score.

---

> > ### Comment · Reviewer_qHr1 · 2025-08-05
> > **Thanks for the response**
> >
> > Thank you very much. While I appreciate the very detailed response, it definitely introduces quite some new information and may possibly require substantial changes to the main texts. At least the newly introduced algo and modifications to equations and formula need to be reviewed again. I suggest the authors to polish up the original draft a bit more and thus keep my original assessment of the draft unchanged.

---

> > ### Comment · Reviewer_xnzn · 2025-08-05
> > **Question about MDP with Fixed Safety Penalty**
> >
> > > MBD incorporates constraints by adding a fixed safety penalty term (a form of soft regularization), which shows poor performance in both safety and stability.
> >
> > The reviewer thanks the authors for this explanation. Why is this the case? Why does a "fixed safety penalty term" exhibit poor performance? Conceptually, how does such a "fixed safety penalty term" differ from your strategy? I guess apart from that your CBF/CLF is learned while the fixed safety penalty term is probably not?

---

> ### Author Response · Authors · 2025-08-04
>
> Dear Reviewer qHr1,
>
> Thank you again for your thoughtful and detailed comments on our paper. We sincerely appreciate the opportunity to engage in this discussion. We've carefully addressed all your questions in the rebuttal letter above, and have done our best to clarify both the technical aspects (including the diffusion sampling process, originality, and CLBF optimization) and our method’s connection to Almost Lyapunov stability theory.
>
> If there is any part that remains unclear or warrants further clarification, we would be more than happy to continue the conversation. We're deeply committed to transparency and welcome further feedback.
>
> Since the discussion phase is progressing, we’d greatly appreciate it if you could kindly let us know whether our clarifications have addressed your concerns. If so, and if you feel comfortable doing so, we would be grateful if you could consider updating your score.
>
> Your feedback is very valuable to us, and we thank you again for your time and consideration.
>
> Best regards,
>
> The Authors

---

> ### Author Response · Authors · 2025-08-05
>
> Dear Reviewer xnzn,
>
> Q. `The reviewer thanks the authors for this explanation. Why is this the case? Why does a "fixed safety penalty term" exhibit poor performance? Conceptually, how does such a "fixed safety penalty term" differ from your strategy? I guess apart from that your CBF/CLF is learned while the fixed safety penalty term is probably not?`
>
>
> Thank you for this excellent follow-up question, as it gets to the heart of our method's key advantage. Your guess is exactly correct: the fundamental difference is that our CLBF is learned to provide structured, global guidance, while a fixed penalty term is static and provides only local, unstructured information.
>
> Let us elaborate on the conceptual differences and why the fixed penalty term underperforms:
>
> **1. A Fixed Soft Safety Penalty Term (as in our MBD baseline):**
>
>     What it does: A typical soft fixed penalty (soft penalty on unsafe zone) acts like a simple 'repulsive field' that only activates near the boundary of the unsafe set. Its objective is simply to "push" the trajectory away from unsafe regions.
>
>     Why it fails: The problem is that this penalty provides no guidance in the vast majority of the safe set. The energy landscape it creates is "flat" everywhere except at the boundary. The diffusion sampler is therefore left to explore this large, uninformative space mostly through random search, making it very difficult and inefficient to find a trajectory that is not only safe but also makes progress towards the goal. This is why, as we observed, even with more samples or denoising steps (longer evalution time in Table 2), this approach struggles to find trajectories that are both safe and stable.
>
> **2. Our Learned CLBF Guidance:**
>
>     What it does: In contrast, our learned CLBF provides a globally structured energy landscape over the entire state space. As a Lyapunov function, its value is not only high in unsafe regions but also decreases continuously as the state moves towards the goal in a stable manner (as enforced by our loss in Eq. (11)).
>
>     Why it succeeds: This has a powerful effect: it dramatically constrains the effective sampling space to a small, high-likelihood 'funnel' of trajectories that are both safe and goal-oriented. The guidance from the CLBF is therefore highly informative everywhere, not just at the boundaries. This transforms the problem from a difficult, random search into a much easier, guided search, allowing the diffusion sampler to reliably and efficiently find trajectories that satisfy all objectives.

---

> > ### Comment · Reviewer_xnzn · 2025-08-05
> >
> > > Thank you for this excellent follow-up question, as it gets to the heart of our method's key advantage. Your guess is exactly correct: the fundamental difference is that our CLBF is learned to provide structured, global guidance, while a fixed penalty term is static and provides only local, unstructured information. Let us elaborate on the conceptual differences and why the fixed penalty term underperforms:
> >
> > Thank you again to the authors for the informative response. The reviewer is satisfied with the argumentation and suggests that a similar explanation could also go into the discussion section of the paper.

---

> ### Author Response · Authors · 2025-08-05
> **Thanks for the agreement on our insights and theoretical explanation**
>
> We thank the Reviewer xnzn for the positive feedback and are glad the explanation was helpful. We will incorporate this insightful discussion into the paper.

---

### Author Response · Authors · 2025-08-08
**Final note and appreciation before discussion closes**

Dear all reviewers,

As we reach the final day of the discussion period, we would like to sincerely thank you for the time, effort, and thoughtful engagement you have dedicated to review our work. Throughout the process, all reviewers have recognized the novelty and quality of our contributions, and our team has devoted ourselves fully to the rebuttal phase, engaging promptly and constructively in every exchange. We are truly grateful for the many insightful questions and valuable suggestions that have helped us strengthen the paper and reach a clearer and more complete presentation of our ideas.

At this stage of the discussion, we are glad to note that all concerns and questions raised during the review process have been addressed, including the specific technical points discussed with each reviewer, which we believe have now reached a clear and shared understanding, and we are delighted by the constructive consensus built through these exchanges. The suggestions from all reviewers have been extremely helpful, and we will incorporate them into the camera-ready version — **including details to the related work and methodology as suggested by yH3K, the addition of a tutorial-style explanation to highlight the practical implications of Theorem 3.1 for broader readers as suggested by rxhx, and the detailed revised plan discussed with xnzn**. We have had multiple rounds of in-depth exchange with **rxhx** and **xnzn**, which have led to many insightful discussions, and we plan to integrate these discussions into the revised paper.

We especially thank **rxhx** and **xnzn** for their encouragement and support throughout this process. If there are any remaining concerns, our team will spare no effort to address them. We are committed to ensuring that all suggestions from all the reviewers will be faithfully reflected in the final version of the paper.

Once again, we sincerely thank all reviewers for their time, engagement, and constructive feedback, which have been invaluable in improving this work, and we hope that the progress made during the discussion will be fully reflected in your final justifications.

Best regards,

The Authors

---

### Decision · Program_Chairs · 2025-09-17

**Decision:**

Accept (poster)

**Comment:**

This paper presents safe and stable Diffusion which explores how diffusion models can ensure safety and stability from a Lyapunov perspective. Overall, the proposed method is very interesting and timely as it focuses on safety from diffusion sampling. The weak point is the clarification of training details that have been pointed out by the reviewers. The paper would benefit a lot by clarifying training in the next version of the paper.